# TicketLLM: Next-Generation Sparse and Low-bit Transformers with Supermask-based Method

**Yasuyuki Okoshi**                                          *okoshi.yasuyuki@artic.iir.isct.ac.jp*
*AI Computing Research Unit*
*Institute of Science Tokyo*

**Hikari Otsuka**                                              *otsuka.hikari@artic.iir.isct.ac.jp*
*AI Computing Research Unit*
*Institute of Science Tokyo*

**Daichi Fujiki**                                                 *dfujiki@artic.iir.isct.ac.jp*
*AI Computing Research Unit*
*Institute of Science Tokyo*

**Masato Motomura**                                         *motomura@artic.iir.isct.ac.jp*
*AI Computing Research Unit*
*Institute of Science Tokyo*

**Reviewed on OpenReview:** *https://openreview.net/forum?id=sE69HKykQw*

## Abstract

Strong Lottery Tickets (SLTs) are subnetworks within a randomly weighted network uncovered by a binary mask called supermask. They offer a promising approach to model compression by eliminating the need to store weights since their effective subnetwork can be regenerated from a fixed random seed and the supermask. However, extending this approach to large language models (LLMs) is non-trivial due to limited scalability and inefficient training dynamics of existing SLT methods. To address these challenges, we propose Adaptive Supermask (Ada-Sup), a scalable and efficient method for discovering high-quality multi-bit supermasks through an innovative quantization-based approach. Building on this method, we introduce TicketLLM, a low-bit and sparse Transformer-based LLM architecture powered by Ada-Sup. Experimental results show that Ada-Sup can discover high-quality supermasks with significantly reduced training costs compared to previous methods in both binary and multi-bit settings. Furthermore, TicketLLM outperforms BitNet b1.58 on a 1.3B parameter model with the same memory per connection, achieving 0.6% reduction in perplexity (from 13.62 to 13.54) while operating at a higher sparsity level (around 50% vs. around 33%). These results highlight the potential of supermask-based methods as a promising approach for building lightweight LLMs. Code is available: `https://github.com/yasu0001/TicketLLM`.

## 1 Introduction

As the number of parameters in Transformer-based large language models (LLMs) grows, model compression techniques such as pruning and quantization have become critical for efficient deployment. To achieve a better balance between compression ratio and model performance, optimization methods that incorporate compression directly into model training have gained increasing interest (Wang et al., 2023; Ma et al., 2024; Fang et al., 2024; Liu et al., 2025). In particular, BitNet papers (Wang et al., 2023; Ma et al., 2024) analyze scaling behavior of low-bit LLMs trained from scratch, uncovers novel efficiency characteristics, revealing scaling laws that offer advantages over LLMs with higher precision. Unlike conventional quantization or pruning methods that are applied to pre-trained models, these works investigate training directly with low-bit weights. As a result, the primary research focus has shifted toward discovering low-bit representations that are highly compatible with the pre-training. However, existing approaches have primarily focused on quantization and sparsity to achieve efficient low-bit representations, leaving other promising approaches underexplored.

Table 1: Comparison of TicketLLM with other reproduced LLMs with 1.3B parameters. All models are trained with approximately 400B tokens with the same dataset sampled from FineWeb-Edu (Penedo et al., 2024). LLaMA* is configured LLaMA architecture by replacing linear projection layers with low-bit representations with RMSNorm.

| Name | TicketLLM | BitNet b1.58 | LLaMA |
|---|---|---|---|
| *General Characteristics* | | | |
| Base Arch. | LLaMA* | LLaMA* | LLaMA |
| Weight Distribution | {-3, -2, ..., 3} | {-1, 0, +1} | bf16 |
| Compression Approach | SLT | Quantization | N/A |
| Training Method | Ada-Sup | QAT | N/A |
| *Parameters* | | | |
| Random Weights | {-1, +1} | N/A | N/A |
| Trained Weights | N/A | {-1, 0, +1} | bf16 |
| Supermask | 2-bit ({0,1,2,3}) | N/A | N/A |
| *Memory and Sparsity* | | | |
| Memory per Connection | 2-bit | 2-bit | 16-bit |
| Measured Sparsity (%) | **≈ 50** | ≈ 30 | N/A |
| *Performance* | | | |
| C4 PPL (1.3B) | **13.54** | 13.62 | 11.68 |

The Strong Lottery Tickets (SLTs) (Zhou et al., 2019; Ramanujan et al., 2020) and their follow-up study (Hirose et al., 2022) have introduced a novel compression paradigm that incorporates randomness while preserving the advantages of conventional pruning and quantization. The core insight of SLTs is that effective sparse neural networks can be discovered within a randomly weighted neural network using connectivity masks called supermasks. Since random numbers can be generated with a random number generator from a pre-defined seed on the fly, such randomness in SLT can eliminate the need for memory access for weights. Furthermore, SLTs offer similar benefits to pruning and quantization through a fundamentally different mechanism: identifying the sparse subnetwork and storing only a binary supermask and a seed. This novel paradigm has sparked considerable interest and found applications in diverse domains, including Graph Neural Networks (Huang et al., 2022), Folded Networks (García-Arias et al., 2023), one-layer Transformers (Shen et al., 2021), and large-scale vision applications (Okoshi et al., 2022).

Despite its potential, the impact of SLTs on the scaling behavior of Transformers remains largely unexplored, primarily due to the lack of suitable optimization algorithms. Specifically, there are two major challenges in applying SLTs to LLMs: (1) ensuring scalability to complex tasks and (2) reducing the training overhead associated with supermask optimization. Multicoated Supermasks (M-Sup) (Okoshi et al., 2022) address the scalability issue by introducing a multivalued mask, accumulating multiple supermasks with different sparsity to represent the importance scale of random weights. However, it still suffers from training inefficiencies, mainly due to overhead for supermask generation that maps importance score parameters assigned to each connection into supermask.

To tackle these challenges, we propose Adaptive Supermasks (Ada-Sup), a scalable and efficient supermask optimization method based on a quantization-based approach applied to score parameters. As the bit precision of quantization can be flexibly adjusted to balance model efficiency and performance, quantization enables multivalued mask extension without introducing additional training complexity. This approach is also computationally efficient as quantization mostly consists of element-wise operations such as scaling, clipping, and rounding. We then introduce TicketLLMs, a family of Transformer-based models optimized with Ada-Sup, to evaluate how the proposed method scales in large language models. TicketLLMs are designed based on the LLaMA-based architecture with the additional normalization layers for inputs of linear projections as proposed by Ma et al. (2024). We configure four model variants ranging from 110M to 1.3B parameters, trained on up to 416B tokens.

First, we compare Ada-Sup with existing supermask optimization methods for Transformer architectures. Subsequently, we evaluate TicketLLMs across various amounts of training tokens and parameters to assess their scalability and compare them with BitNet b1.58, a state-of-the-art ternary LLM trained from scratch. Our experimental findings are summarized as follows:

- Compared with other supermask optimization methods, Ada-Sup achieves comparable performance with lower training costs in binary and multi-valued supermask settings.

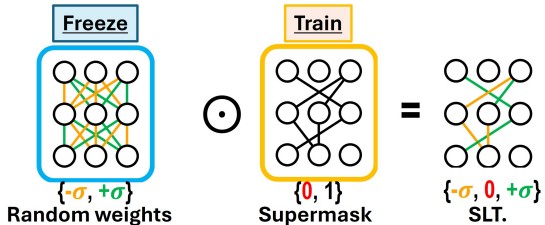

Figure 1: Structure of Strong Lottery Ticket (SLT). Since random weights are sampled from Signed Kaiming Constant, the distribution of subnetwork becomes ternary values.

- Among the methods that leverage unstructured sparsity in low-bit representations, allocating additional bits to the supermask is more effective than using them for sign representation.
- As summarized in Table 1, experiments on the 1.3B parameters show that our proposal improves perplexity by 0.08 even with higher sparsity compared with BitNet b1.58, using 2-bits per connection.

This paper is the first work to comprehensively analyze the scaling behavior of SLT in LLMs, opening new possibilities for efficient Transformer architectures.

## 2 Strong Lottery Ticket

### 2.1 Formulation of SLT

As shown in Figure 1, SLT consists of random weights and a supermask. During inference, a sparse subnetwork is computed with the element-wise product of the random weights and the supermask. Given an input row vector $\mathbf{x} \in \mathbb{R}^{1 \times C_{\text{in}}}$, an output row vector $\mathbf{y} \in \mathbb{R}^{1 \times C_{\text{out}}}$, and random weights $\mathbf{W}_{\text{rand}} \in \mathbb{A}^{C_{\text{in}} \times C_{\text{out}}}$, where $\mathbb{A}$ is an arbitrary distribution, a linear projection of SLT with a batch size of 1 is defined as follows:

$$\mathbf{y} = \mathbf{x} \left( \mathbf{W}_{\text{rand}} \odot \mathbf{M} \right), \tag{1}$$

where $\mathbf{M} \in \{0,1\}^{C_{\text{in}} \times C_{\text{out}}}$ is a supermask that uncovers the subnetwork of random weights.

The choice of the random weight distribution $\mathbb{A}$ plays a critical role in SLT. Ramanujan et al. (2020) explored two variants of distributions: Kaiming Normal $\mathcal{N}(0, \sigma)$ and Signed Kaiming Constant (SKC), a binary distribution over $\{-\sigma, \sigma\}$, where $\sigma$ is the standard deviation of the Kaiming Normal distribution. Among them, the SKC not only offers better performance but also improves computational efficiency by leveraging efficient operations enabled by its binary representation. Following this observation, we adopt the binary distribution in this work.

Another important aspect of SLT is discovering an effective supermask. Since the supermask takes only binary values (0 or 1), its optimization is known as NP-hard. Most prior work introduces score parameters $\mathbf{S} \in \mathbb{R}_{\geq 0}^{C_{\text{in}} \times C_{\text{out}}}$ to make the optimization tractable. These scores are then mapped to the supermask. Accordingly, Eq. 3 can be rewritten using a mapping function $\mathcal{F} : \mathbb{R}^{C_{\text{in}} \times C_{\text{out}}} \to \{0,1\}^{C_{\text{in}} \times C_{\text{out}}}$ as:

$$\mathbf{y} = \mathbf{x} \left( \mathbf{W}_{\text{rand}} \odot \mathcal{F}(\mathbf{S}) \right). \tag{2}$$

Various strategies have been proposed to find better functions $\mathcal{F}$. Below, we briefly describe three representative approaches: EdgePopup, ProbMask, and FixedTh. EdgePopup (Ramanujan et al., 2020) is the first to establish a practical framework for finding SLTs. It maps the top-$k\%$ of scores to 1 while mapping other scores to 0. Thus, function $\mathcal{F}$ can be rewritten with the hyperparameter for supermask density $k$ as $\mathcal{F}(\mathbf{S}; k)$. ProbMask (Zhou et al., 2021) introduces Gumbel-Softmax mapping to improve the model performance in highly sparse regions. FixedTh (Koster et al., 2022) compares scores to a threshold hyperparameter $\tau$ to calculate supermask, setting $M_{i,j} = 1$ if $S_{i,j} \geq \tau$, and 0 otherwise. While FixedTh introduces sign flipping in addition to supermask training, experiments in this paper turned it off to leverage random weights for storage savings. These methods update scores by approximating gradient of $\mathcal{F}$ such as straight-through estimators (Bengio et al., 2013) during training.

Okoshi et al. (2022) have proposed Multicoated Supermask (M-Sup), which expands EdgePopup to represent varying levels of importance using multivalued mask. This method expands to multivalued mask by stacking

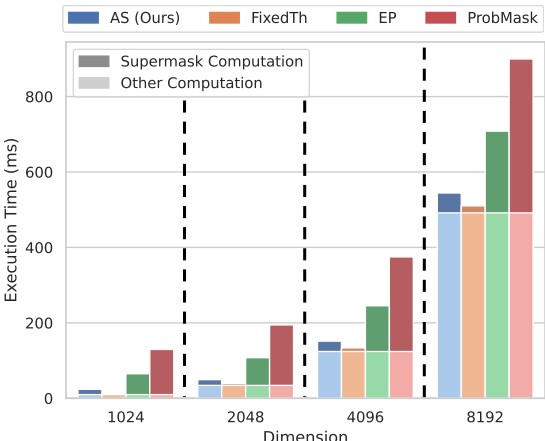

Figure 2: Overhead of supermask optimization from score parameters for each method. We include three baselines: EP (Ramanujan et al., 2020), ProbMask (Zhou et al., 2021), and FixedTh (Koster et al., 2022), in addition to our proposa (AS). We set the same input and output dimensions with all methods with a batch size of 2048. Execution time is measured over 100 iterations using an NVIDIA GeForce RTX 3090. Execution time is measured using the native PyTorch implementation (without custom CUDA kernels or third-party optimizations).

supermasks with different sparsity, represented as:

$$\mathbf{y} = \mathbf{x} \left( \mathbf{W}_{\text{rand}} \odot \sum_{i=1}^{N} \mathcal{F}(\mathbf{S}; k_i) \right), \tag{3}$$

where $k_1, \ldots, k_N$ are density hyperparameters corresponding to each supermask. Since different supermasks share the same score parameters, the $(i+1)$-th supermask becomes a subset of the $i$-th supermask. This method has successfully scaled to complex computer vision tasks.

## 2.2 Model Compression View of SLT

The unique property of SLTs, where only connectivity is learned, offers an attractive paradigm for model compression, particularly in resource-constrained environments. This subsection briefly discusses SLT from the perspective of model compression.

SLTs can be interpreted as a special case of a concurrent blend of pruning and quantization. Specifically, SLTs sparsify the model by uncovering the subnetwork through the supermask, which determines the connectivity of random weights. This sparsity has a similar granularity to the unstructured sparsity in conventional pruning methods. Unlike conventional quantization, SLT achieves low-bit storage and computation through a fundamentally different mechanism. Storage efficiency is obtained by leveraging a binary supermask and an appropriate architectural support for inference-time random weight generation, such as the one found in Hirose et al. (2022), eliminating the need for explicit weight storage. Low-bit computation is enabled by sampling random weights from a binary distribution, which empirically yields optimal performance in SLT training (Ramanujan et al., 2020). Consequently, SLT can effectively represent a ternary weighted model at the memory requirements of only the binary supermask.

In the case of a multivalued mask extension, SLT-based models retain the benefits of sparse and low-bit representations while gaining additional model expressiveness. Due to the randomness of SLT-based models, an $(n+1)$-bit symmetric quantized representation can be encoded using only $n$ bits in a multimask, resulting in improved storage efficiency, especially for extremely low-bit representations.

## 2.3 Limitations of Existing SLT Frameworks

While SLT-based models are promising candidates for efficient neural networks, existing frameworks face critical scalability and training efficiency challenges when applied to LLMs. Conventional SLT frameworks typically focus on optimizing a single binary supermask, which restricts the search space to binary connectivity

patterns. This limited expressiveness often leads to performance degradation on complex tasks, as Okoshi et al. (2022) observed. Increasing the number of supermasks is a natural way to enhance model capacity, but implementing such modifications is non-trivial.

M-Sup has successfully increased the model capacity of SLT-based models by expanding EdgePopup to represent multivalued masks. Such extension necessitates extra sparsity hyperparameters, increasing the complexity of training. Although the paper reduces increased hyperparameters with two strategies, the accurate one relies on the pre-trained scores of the binary supermask, which is impractical in LLMs since multiple pre-training runs are required for one model.

Another challenge lies in the computational overhead associated with supermask generation. As shown in Figure 2, computing supermask during training introduces non-negligible overhead. Specifically, EdgePopup incurs between 30.6% and 85.0% of the total linear projection time, the primary computation in LLMs, for supermask computation. While FixedTh offers a highly efficient supermask optimization due to its simplicity, it still encounters similar limitations to M-Sup when extending it to multivalued mask.

These observations highlight a gap in existing SLT-based methods: achieving both model scalability and training efficiency remains challenging. To address this difficulty, we propose Adaptive Supermask (Ada-Sup), a novel supermask optimization framework that improves both scalability and efficiency through quantized score-based mask generation.

## 3 Adaptive Supermask and Its Application to LLMs

### 3.1 Adaptive Supermask

As discussed in the previous section, extending SLT frameworks for large language models must reduce supermask generation overhead while ensuring scalability without additional complexity for multivalued mask. Adaptive Supermasks (Ada-Sup) tackle these challenges with a quantization-based approach for score parameters.

The brief overview of our proposal is described in Figure 3. As discussed in Sec. 2.1, supermask is optimized by updating score parameters. Given score parameters $\mathbf{S} \in \mathbb{R}_+^{C_{in} \times C_{out}}$, Ada-Sup calculates supermask by quantizing score parameters, as

$$\mathcal{F}(\mathbf{S}) = \gamma \lceil \text{clip}(\mathbf{S}/\gamma, 0, 1) \rfloor. \tag{4}$$

Here, $\gamma$ is a scaling factor that determines the clip range of scores, and $\lceil \cdot \rfloor$ is the round function. The $\text{clip}(x, a, b)$ function clamps all elements to the range $[a, b]$. Please note that the supermask in Ada-Sup uses a scaled binary mask $\mathbf{M} \in \{0, \gamma\}$, which is different from the original supermask by introducing the quantization scaling factor.

Ada-Sup can extend to a multivalued mask with $n$-bits representation by replacing the upper bound of the clip function with $2^n - 1$ without additional operations:

$$\mathcal{F}(\mathbf{S})_{\text{multi}} = \gamma \lceil \text{clip}(\mathbf{S}/\gamma, 0, 2^n - 1) \rfloor. \tag{5}$$

In the backward pass, we use the straight through estimators (Bengio et al., 2013) to compute the derivative of $\mathcal{F}$ concerning the score $\mathbf{S}$, as in previous methods.

Weight quantization often determines the scaling factor $\gamma$ based on the distribution of weights. Following BitNet b1.58 (Ma et al., 2024), we compute $\gamma$ as the mean of the absolute values of the parameters. Since score parameters are always non-negative, we can directly use their mean without taking absolute values:

$$\gamma = \frac{1}{MN} \sum_{i,j} |S_{ij}| = \frac{1}{MN} \sum_{i,j} S_{ij}. \tag{6}$$

### 3.2 Weight Initialization

In order to find accurate SLTs, the distribution of random weights is crucial. Previous research (Ramanujan et al., 2020; Okoshi et al., 2022) has demonstrated that the Signed Kaiming Constant (SKC), which samples from $\{-\sigma, \sigma\}$, yields the best performance ($\sigma$ is the standard deviation of the Kaiming Normal

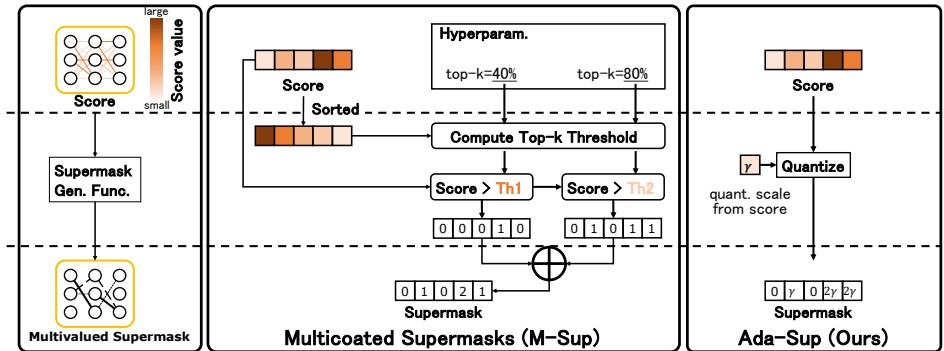

Figure 3: Overview of supermask generation methods. Supermask including both single and multivalued supermasks is generally optimized using score parameter assigned to each connectivity, based on supermask generation function (left). M-Sup (Okoshi et al., 2022), a fundamental approach for multivalued mask, calculates supermask by selecting top-k% of score parameters, resulting in significant training overhead (center). Ada-Sup, in contrast, reduces computational overhead through score quantization-based approach (right).

distribution (Han et al., 2015)). However, since the supermasks are already scaled in our method, as shown in Eq. 4, multiplying $\gamma$ with $\sigma_{KN}$ introduces a redundant operation. Therefore, we adopt a binary distribution $\{-1, +1\}$ for random weights.

### 3.3 Overall Architecture of TicketLLM

This section presents overall architecture of TicketLLM, which is designed to showcase the effectiveness of Ada-Sup on LLMs. We follow the LLaMA architecture (Touvron et al., 2023), including rotary positional embedding (RoPE) (Su et al., 2024) and gated linear unit (GLU) (Shazeer, 2020). The basic block of LLaMA consists of a multi-head attention (MHA) block and a feed-forward network (FFN) block with a residual connection. Different from the LLaMA architectures, we eliminate the pre-normalization layer for MHA and FFN since we introduce the RMSNorm to the input of the linear projection layer following Ma et al. (2024). Thus, the output of a Transformer block is calculated as follows:

$$
\begin{aligned}
\mathbf{X}_{\text{mid}} &= \mathrm{X}_{\text{in}} + \mathrm{MHA}(\mathbf{X}_{\text{in}}), \\
\mathbf{X}_{\text{out}} &= \mathrm{X}_{\text{mid}} + \mathrm{FFN}(\mathbf{X}_{\text{mid}}).
\end{aligned}
$$

Here, $\mathbf{X}_{\text{in}}, \mathbf{X}_{\text{mid}}, \mathbf{X}_{\text{out}} \in \mathbb{R}^{T \times d}$ denote input sequence, output sequence of the MHA, and output sequence of FFN, respectively, where $T$ is the sequence length and $d$ is the model dimension.

TicketLLM replaces all linear projections in both MHA and FFN with Ada-Sup linear (ASL). Thus, given an input sequence $\mathbf{X}_{\text{in}}$, the MHA layer can be represented as:

$$
\mathrm{MHA}(\mathbf{X}_{\text{in}}) = \phi \left( \frac{\mathrm{ASL}_{\mathrm{Q}}(\mathbf{X}_{\text{in}}) \left( \mathrm{ASL}_{\mathrm{K}}(\mathbf{X}_{\text{in}}) \right)^{\mathrm{T}}}{\sqrt{d}} \right) \mathrm{ASL}_{\mathrm{V}}(\mathbf{X}_{\text{in}}),
$$

where $\phi$ is a softmax function, and $\mathrm{ASL}_{\mathrm{Q}}, \mathrm{ASL}_{\mathrm{K}}$, and $\mathrm{ASL}_{\mathrm{V}}$ are query, key, and value projections by Ada-Sup, respectively. Note that we omit the RoPE and assume the single-head attention for simplicity.

Based on the LLaMA architectures, we apply the FFN with GLU. Thus, $\mathbf{X}_{\text{out}} = \mathrm{FFN}(\mathbf{X}_{\text{in}})$ is calculated using the following two steps:

$$
\begin{aligned}
\mathbf{X}_{\text{mid2}} &= \mathrm{ASL}_1(\mathbf{X}_{\text{mid}}) \odot \sigma(\mathrm{ASL}_2(\mathbf{X}_{\text{mid}})) \\
\mathbf{X}_{\text{out}} &= \mathrm{ASL}_3(\mathbf{X}_{\text{mid2}}),
\end{aligned}
$$

where $\sigma$ represents the sigmoid function. As an exception, we do not apply Ada-Sup to the weights shared between the token embedding and the final linear projection.

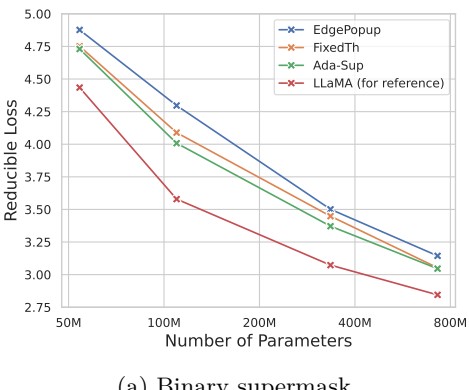 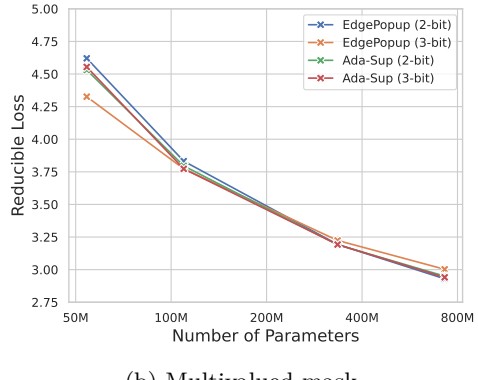

(a) Binary supermask

(b) Multivalued mask

Figure 4: Comparison of Ada-Sup with other supermask methods, including EdgePopup and FixedTh. To determine the $\gamma$ in Eq. 4, we use the mean of scores for Ada-Sup. We also include the conventional weight learning with BF16 (LLaMA) as a reference. We vary the number of parameters from 0.05B to 0.7B with a fixed TPP of 20.

## 4 Evaluation

### 4.1 Experimental Setup

Transformer models are trained on randomly sampled subsets from FineWeb-Edu (Penedo et al., 2024) and evaluated on C4 validation dataset (Raffel et al., 2020). Both datasets are tokenized using the LLaMA2 tokenizer (Touvron et al., 2023), whose vocabulary size is $32K$. In order to ensure consistent training, tokens are concatenated into sequences of length 2048, where shorter sequences are combined, and longer sequences are truncated.

We vary model parameters from 0.05B to 1.3B by increasing the number of layers and hidden dimensions while keeping the head dimension constant. Table 4 provides the detailed model configurations. Scores are initialized using a normal distribution with a standard deviation of 0.02. We increase the number of training tokens for pre-training following a ratio of **t**okens **p**er model **p**arameters (TPP). Models are optimized with decoupled weight decay (AdamW) (Loshchilov & Hutter, 2019), setting $\beta_1 = 0.95$, $\beta_2 = 0.99$, and a weight decay of 0.1. The maximum learning rate is scaled down with increasing parameters, according to Kaplan et al. (2020). It linearly decays to zero after the learning rate warms up in the first 1% of the total number of iterations. Although cosine decay is commonly used for pre-training, we adopt linear decay scheduling following the recent findings (Defazio et al., 2024; Anonymous, 2025). The batch size is 512, with gradient accumulation employed for larger models. Gradient clipping with 1.0 is also applied to stabilize training. All hyperparameters are summarized in Table 5.

In addition to cross-entropy loss, we evaluate perplexity and downstream accuracy for some experiments to provide a more comprehensive analysis of model performance in practical situations. Downstream tasks includes 0-shot performance on ARC-easy, ARC-challenge (Yadav et al., 2019), HellaSwarg (Zellers et al., 2019), BoolQ (Clark et al., 2019), Open-bookQA (Mihaylov et al., 2018), PIQA (Bisk et al., 2020), WinoGrande (Sakaguchi et al., 2021), COPA (Roemmele et al., 2011), MMLU (Hendrycks et al., 2021), and LAMBADA (Paperno et al., 2016). Validation is performed once the training is completed using the latest checkpoint. All training and evaluation experiments are conducted using the LLM Foundry (MosaicML, 2023).

### 4.2 Comparison of Supermask Methods

This section compares Ada-Sup with other supermask methods, including EdgePopup and FixedTh. We omit ProbMask from the comparison due to its substantial overhead in supermask computation. We also include reproduced LLaMA model results as a baseline. Following Ramanujan et al. (2020) and Koster et al. (2022), we set the sparsity to 50% for EdgePopup and use a fixed threshold of 0.01 for FixedTh, respectively. We evaluate transformer models ranging from 0.05B to 0.7B parameters trained with a fixed TPP of 20. Evaluation is performed using cross-entropy loss on the C4 validation dataset.

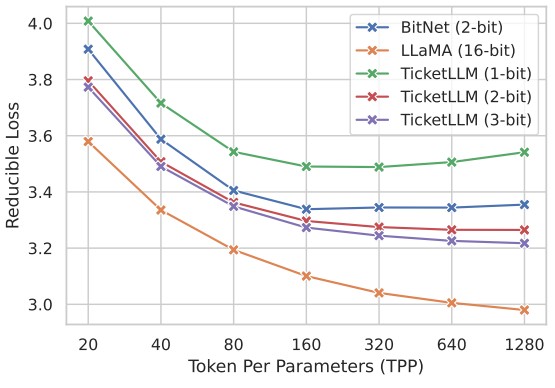 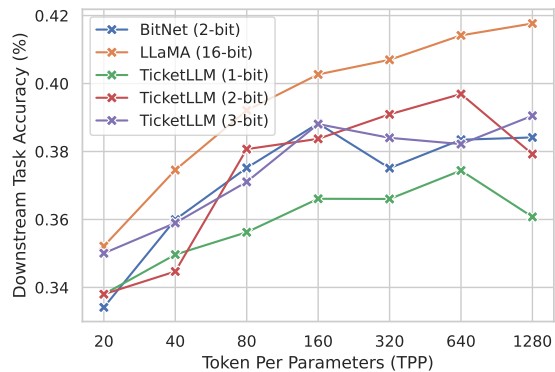

(a) Evaluation loss on C4-validation dataset      (b) Average accuracy on 10 downstream tasks

Figure 5: Model performance scaling regarding training tokens for different numerical representations with 0.1B parameters.

Figure 4 (a) shows that Ada-Sup consistently outperforms other baselines except for LLaMA models across all parameters in the binary supermask setting. Specifically, Ada-Sup reduces the loss by 0.05 compared to FixedTh and by 0.17 compared to EdgePopup on average, demonstrating that Ada-Sup discovers superior supermasks compared to other methods.

Figure 4 (b) compares Ada-Sup against M-Sup with different supermask bits. M-Sup is a multivalued mask optimization method where each supermask is optimized using EdgePopup. To align overall training iterations with our method, sparsities for each supermask in M-Sup are determined using a uniform setting. As shown in the figure, both methods achieve comparable performance. For example, Ada-Sup achieves an evaluation loss of 2.95 compared to 2.92 from EdgePopup for 2-bit supermasks, while for 3-bit supermasks, the losses are 2.94 and 3.00, respectively. Despite comparable performance, Ada-Sup shows superior training efficiency. On 700M-parameter models trained with 20 TPPs, Ada-Sup takes a training time of approximately 40 H100 GPU hours for both 2-bit and 3-bit supermasks, while M-Sup with EdgePopup takes around 80 H100 GPU hours.

These results highlight that Ada-Sup not only matches performance in both the binary and multivalued mask settings but also significantly reduces the computational cost in multivalued mask methods, making it a more practical and scalable solution for applying supermask-based training to LLMs.

### 4.3  Analysis of Scaling Laws

### Dataset Scaling

This section explores how increasing the number of training tokens affects model performance across different low-bit sparse representations. To analyze this dataset scaling, we compare the loss on the C4 validation dataset across a wide range of TPPs, from 20 to 1280, under a fixed model size of 0.1B parameters. We include BitNet b1.58 (Ma et al., 2024) as a strong low-bit sparse baseline, which adopts ternary representations as weights. We also reproduce BF16 LLaMA-based models as the dense baselines. Results are summarized in Figure 5. While TicketLLM with a 1-bit supermask represents the binary supermask setting, the 2-bit and 3-bit variants employ the multivalued mask extension.

We observe three key findings from Figure 5 (a). (1) When comparing TicketLLM models with different supermask bits, increasing supermask consistently improves performance. However, the performance gain from 2-bits to 3-bits is relatively small compared to the improvement from 1-bit to 2-bits. Thus, TicketLLM with 2-bit is the best choice, considering the model size increase from 2-bit to 3-bit. Based on this observation, we use 2-bit TicketLLMs in the following experiments.

(2) TicketLLM not only outperforms ternary quantization but also shows better scaling trends regarding the dataset size under the same 2-bit model. While BitNet-b1.58 does not improve its performance beyond

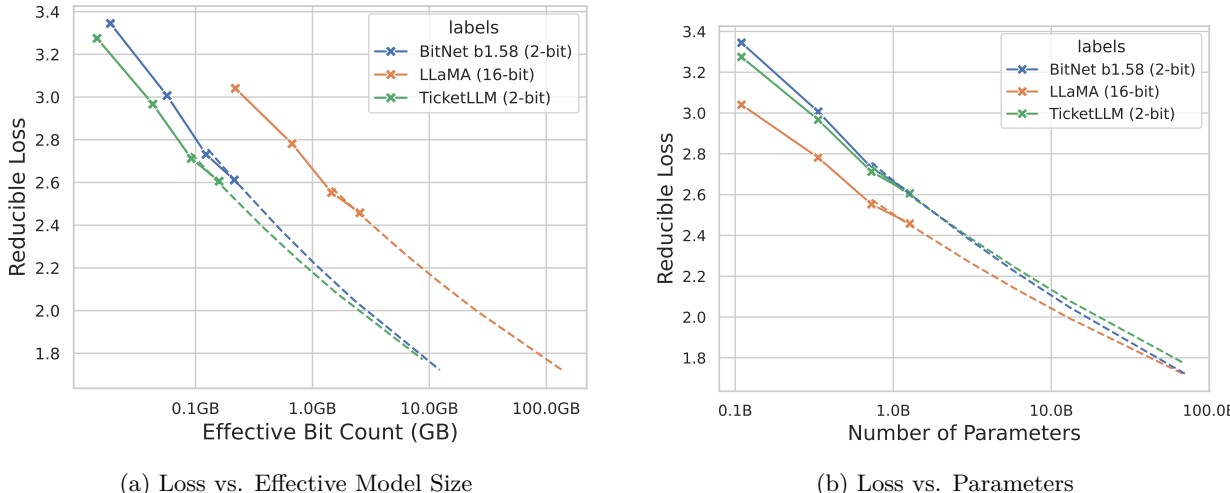

(a) Loss vs. Effective Model Size (b) Loss vs. Parameters

Figure 6: Scaling trends of reducible loss concerning (a) effective bit count and (b) the number of parameters. Effective bit count is calculated as the product of the number of parameters, bits per parameter, and sparsity ratio (1 - sparsity). Solid lines represent actual evaluation results on C4 validation set, while dashed lines indicate extrapolated values fitted following the assumption in Gadre et al. (2024). Both TicketLLM and BitNet b1.58 require 2-bit per parameter.

TPP=160, TicketLLM continues to benefit from the increased dataset. This result comes from the extended search space of TicketLLM, which fully leverages 2-bit representation and expanded numerical representation.

(3) When comparing TicketLLM with a LLaMA-based dense model, on the other hand, the loss gap gradually widens as the number of training tokens increases. This result suggests that improving model capacity to handle increasing data under low-bit representation remains a key challenge, and addressing this limitation is a promising direction for future research. These results highlight the effectiveness of our proposal, particularly in low-bit settings with larger datasets.

However, the trend differs in downstream tasks, as shown in Figure 5 (b), which reports average accuracy across 10 downstream tasks described in Sec. 4.1. Despite reproduced LLaMA models consistently improving accuracy with more training data, low-bit models, including TicketLLM and BitNet b1.58 exhibit a weaker correlation between training tokens and downstream performance. This result indicates a current limitation of low-bit models in acquiring transferable knowledge from large-scale data. Although the difference in downstream accuracy between TicketLLM with 2-bit and BitNet b1.58 is smaller than the corresponding loss difference on the validation set, it still outperforms BitNet b1.58 in larger data where the TPP exceeds 320, except at TPP = 1280. These findings reinforce insights from the loss analysis and highlight the challenges in designing and optimizing low-bit models, particularly for downstream generalization.

**Parameter Scaling**

This section investigates the scaling behavior of TicketLLMs as the number of parameters increases. We compare the performance of 2-bit TicketLLM with BitNet b1.58 (2-bit) and LLaMA (16-bit) across a wide range of model parameters from 0.1B to 1.3B. All models are trained with a fixed TPP of 320. Evaluation is conducted on C4-validation. Effective bit count is calculated as the product of non-zero parameters and bits. While this bit count does not reflect the actual model size due to the overhead of sparse encoding schemes such as CSR, it serves as a useful metric for comparing the theoretical storage requirements across different quantization methods.

We also fit the scaling curve for all models under the assumption in Gadre et al. (2024). The original scaling law formulation assumes

$$L(N, D) = E + AN^{-\alpha} + BD^{-\beta}, \tag{7}$$

Table 2: Perplexity on C4-validation datasets and 0-shot evaluations on 10 downstream tasks. To support a wide variety of downstream tasks, we choose the 0-shot tasks from Gadre et al. (2024) in addition to reported tasks in BitNet b1.58 (Ma et al., 2024) (denoted as BitNet in this table). All models are trained with 320 TPP. Sps. and PPL denote sparsity and perplexity.

| Param. | Model | #Bit | Sps. (%) | PPL ↓ | ARCe | ARCc | HS | OQ | BQ | PQ | WGe | COPA | MMLU | LLAMB | Avg. (%) ↑ |
|---|---|---|---|---|---|---|---|---|---|---|---|---|---|---|---|
| 0.1B | LLaMA | 16 | 0 | 20.92 | 41.33 | 26.11 | 32.16 | 31.60 | 59.39 | 62.57 | 50.36 | 52.00 | 23.17 | 28.29 | 40.70 |
| 0.1B | BitNet | 2 | 32 | 28.35 | 39.94 | 21.76 | 29.17 | 30.20 | 54.25 | 58.22 | 51.30 | 45.00 | 23.29 | 21.99 | 37.51 |
| 0.1B | TicketLLM | 2 | 48 | **26.44** | 37.96 | 23.72 | 28.92 | 30.40 | 60.46 | 60.12 | 50.04 | 55.00 | 23.20 | 21.11 | **39.19** |
| 0.3B | LLaMA | 16 | 0 | 16.14 | 50.29 | 29.61 | 42.53 | 36.20 | 53.43 | 67.14 | 51.93 | 59.00 | 23.17 | 38.09 | 45.34 |
| 0.3B | BitNet | 2 | 32 | 20.22 | 44.40 | 26.88 | 36.01 | 31.20 | 59.45 | 64.36 | 53.28 | 58.00 | 23.38 | 30.18 | 42.51 |
| 0.3B | TicketLLM | 2 | 48 | **19.43** | 43.73 | 25.85 | 36.14 | 34.00 | 58.04 | 63.17 | 50.04 | 58.00 | 23.29 | 30.22 | **42.65** |
| 0.7B | LLaMA | 16 | 0 | 12.84 | 57.95 | 34.39 | 53.81 | 38.40 | 61.13 | 71.27 | 57.85 | 66.00 | 24.37 | 46.90 | 51.81 |
| 0.7B | BitNet | 2 | 32 | 15.35 | 52.78 | 30.63 | 47.44 | 34.40 | 60.49 | 68.39 | 52.96 | 62.00 | 23.31 | 39.84 | **47.42** |
| 0.7B | TicketLLM | 2 | 50 | **15.06** | 49.41 | 29.01 | 45.23 | 34.40 | 56.82 | 67.85 | 54.62 | 64.00 | 23.25 | 37.90 | 46.05 |
| 1.3B | LLaMA | 16 | 0 | 11.68 | 62.92 | 37.29 | 59.87 | 40.20 | 62.26 | 73.01 | 59.67 | 69.00 | 24.18 | 51.52 | 53.99 |
| 1.3B | BitNet | 2 | 33 | 13.62 | 55.39 | 32.68 | 54.51 | 38.40 | 59.24 | 70.35 | 57.30 | 66.00 | 23.80 | 41.78 | **49.75** |
| 1.3B | TicketLLM | 2 | 50 | **13.54** | 54.29 | 33.02 | 52.03 | 36.40 | 59.76 | 70.35 | 54.93 | 66.00 | 23.64 | 42.89 | 49.53 |

where $N$ denotes the number of parameters, $D$ denotes the number of training tokens, and $L(N, D)$ is the cross-entropy loss on the C4 validation set. Following Gadre et al. (2024), we simplify this formulation under two key assumptions: (1) $\alpha = \beta$, and (2) a fixed tokens-per-parameter (TPP) ratio of $D = 320N$. Under these assumptions, the scaling law simplifies to

$$L(N) = E + A' N^{-\alpha}, \tag{8}$$

where $A' = A + 320B$. Thus, we estimate the parameters $E, A'$, and $\alpha$ in this simplified equation. Fitting is performed on models ranging from 0.1B to 1.3B parameters, and the fitted curves are extrapolated to predict performance at larger parameters. Parameter estimation uses Scipy's curve fit function (Virtanen et al., 2020).

Figure 6 presents the results. Solid lines show measured values, while dashed lines denote extrapolated estimates. As shown in the figure, TicketLLMs follow expected scaling laws, with performance improving as parameters increase since the estimated loss exhibits a smooth trend. Such behavior aligns with the principle of scaling laws in Transformer architectures, whose performance scales under fixed TPPs (Hoffmann et al., 2022; Gadre et al., 2024). Compared with BitNet b1.58, TicketLLMs with 2-bit achieve superior loss for effective bit count below 10GB, as shown in Figure 6 (a). While the estimated scaling trends in BitNet b1.58 are slightly better than TicketLLMs, as shown in Figure 6 (b), better trade-offs between effective bit count and performance demonstrates TicketLLMs provide a suitable option in resource constraint devices. With further parameter scaling, the performance gap between BitNet and TicketLLM is expected to remain limited, as both models' performances are expected to be bounded by the full-precision LLaMA baseline. Indeed, the loss gap between LLaMA and 2-bit TicketLLMs narrows with increasing parameters, potentially achieving comparable loss around 100B parameters. This expected convergence suggests that all models would exhibit comparable absolute performance at scale. However, TicketLLM's higher sparsity translates to superior computational efficiency, maintaining its practical advantage in resource-constrained deployments. These results indicate the potential of TicketLLMs as efficient language models. Although validation on larger parameter models is not reported due to resource constraints, we expect similar trends to hold based on the scaling behavior observed in smaller models.

## 4.4 Downstream Evaluations

This section evaluates perplexity on the C4 validation set and 0-shot accuracy across ten downstream tasks over varying model parameters. We use the same models as in the previous section. Table 2 summarizes the results for models ranging from 0.1B to 1.3B parameters.

TicketLLM outperforms BitNet b1.58 across all model sizes, demonstrating both lower perplexity and higher sparsity. While BitNet b1.58 shows superior performance on certain downstream tasks and parameter settings, TicketLLM remains competitive, achieving comparable or even better results in most cases. These findings indicate that TicketLLM is a promising approach to building efficient large language models (LLMs).

Although TicketLLM performs better than BitNet b1.58, it still underperforms LLaMA models in both perplexity and downstream accuracy. Nevertheless, we observe distinct trends in performance scaling across

Table 3: Performance comparison of TicketLLMs with other quantization methods on MobileLLMs with 2-bit storage per parameter. All results for MobileLLMs are cited from Liu et al. (2025).

| Model Name | Method | ARCe | ARCc | BQ | PQ | HS | OBQA | WGe | Avg. (%) ↑ |
|---|---|---|---|---|---|---|---|---|---|
| | FP | **56.0** | **34.5** | **56.3** | **65.5** | **40.1** | **42.2** | **51.3** | **49.4** |
| | RTN | 26.1 | 24.1 | 62.2 | 50.3 | 26.6 | 28.9 | 49.4 | 38.2 |
| | GPTQ | 28.9 | 26.2 | 44.2 | 51.1 | 28.1 | 33.2 | 48.0 | 37.1 |
| MobileLLM-125M | AWQ | 25.8 | 24.2 | 44.2 | 50.7 | 26.2 | 29.2 | 51.6 | 36.0 |
| | OmniQ | 32.4 | 22.7 | 38.1 | 53.4 | 28.2 | 30.9 | 49.9 | 36.5 |
| | SpinQuant | 31.6 | 23.3 | 40.3 | 52.2 | 28.6 | 28.9 | 50.1 | 36.4 |
| | LLM-QAT | 34.9 | 23.3 | 61.8 | 53.8 | 29.1 | 27.4 | 51.3 | 40.2 |
| | ParetoQ | 50.7 | 32.7 | 59.8 | 63.3 | 36.3 | 40.6 | 52.7 | 48.0 |
| TicketLLM-0.1B | Ada-Sup | 38.0 | 23.7 | 60.5 | 60.1 | 28.9 | 30.4 | 50.0 | 41.7 |
| | FP | **65.5** | **42.3** | **57.4** | **71.0** | **53.3** | **47.3** | **58.3** | **56.4** |
| | RTN | 25.9 | 26.5 | 57.1 | 41.0 | 26.6 | 27.3 | 50.3 | 36.4 |
| | GPTQ | 28.6 | 21.5 | 40.5 | 39.2 | 26.6 | 27.3 | 49.0 | 33.2 |
| MobileLLM-350M | AWQ | 27.0 | 23.7 | 45.4 | 49.4 | 26.4 | 28.4 | 50.2 | 35.8 |
| | OmniQ | 33.9 | 29.6 | 54.9 | 58.4 | 31.5 | 32.0 | 52.6 | 41.8 |
| | SpinQuant | 32.4 | 30.5 | 60.0 | 59.1 | 29.6 | 31.9 | 50.1 | 41.9 |
| | LLM-QAT | 40.5 | 25.0 | 49.2 | 56.2 | 34.2 | 34.2 | 52.0 | 41.6 |
| | ParetoQ | 59.0 | 39.3 | 57.4 | 71.0 | 47.2 | 41.7 | 58.2 | 53.4 |
| TicketLLM-0.3B | Ada-Sup | 43.7 | 25.9 | 58.0 | 63.2 | 36.1 | 34.0 | 50.0 | 44.4 |
| | FP | **68.5** | **47.6** | **60.5** | **72.5** | **59.5** | **51.4** | **61.4** | **60.2** |
| | RTN | 25.8 | 26.2 | 37.8 | 37.7 | 25.9 | 26.8 | 50.9 | 33.0 |
| | GPTQ | 29.1 | 25.6 | 48.2 | 49.4 | 27.1 | 27.6 | 52.6 | 37.1 |
| MobileLLM-600M | AWQ | 26.4 | 24.5 | 50.9 | 50.7 | 26.0 | 27.5 | 52.2 | 36.9 |
| | OmniQ | 30.2 | 25.5 | 50.7 | 60.1 | 28.6 | 28.6 | 54.6 | 39.8 |
| | SpinQuant | 32.6 | 24.2 | 39.0 | 58.1 | 29.2 | 30.6 | 49.1 | 37.5 |
| | LLM-QAT | 42.7 | 23.9 | 49.7 | 62.8 | 32.6 | 33.3 | 52.7 | 42.5 |
| | ParetoQ | 68.2 | 47.4 | 64.2 | 73.1 | 58.1 | 50.2 | 62.4 | 60.5 |
| TicketLLM-0.7B | Ada-Sup | 49.4 | 29.0 | 56.8 | 67.9 | 45.2 | 34.4 | 54.6 | 48.2 |
| | FP | **73.9** | **51.4** | **70.0** | **74.8** | **66.4** | **55.1** | **63.2** | **65.0** |
| | RTN | 27.3 | 26.5 | 70.4 | 49.2 | 26.0 | 27.4 | 48.6 | 39.3 |
| | GPTQ | 29.8 | 22.3 | 45.3 | 53.3 | 27.1 | 27.4 | 51.4 | 36.7 |
| MobileLLM-1.5B | AWQ | 30.2 | 26.5 | 54.0 | 59.7 | 28.0 | 29.6 | 52.2 | 40.0 |
| | OmniQ | 36.5 | 24.1 | 59.6 | 60.1 | 30.2 | 32.1 | 52.4 | 42.1 |
| | SpinQuant | 33.3 | 21.5 | 45.3 | 60.5 | 30.1 | 31.2 | 50.3 | 38.9 |
| | LLM-QAT | 41.6 | 25.6 | 50.6 | 61.4 | 31.6 | 33.3 | 50.9 | 42.1 |
| | ParetoQ | 73.3 | 47.5 | 70.1 | 74.1 | 64.6 | 55.9 | 62.5 | 64.0 |
| TicketLLM-1.3B | Ada-Sup | 54.3 | 33.0 | 59.8 | 70.4 | 52.0 | 36.4 | 54.9 | 51.5 |

perplexity and downstream tasks. The performance gap between TicketLLM and LLaMA on perplexity narrows as the number of parameters increases, suggesting improved text generation quality at larger parameters. In contrast, the performance gap on downstream tasks widens with increasing parameters, implying that TicketLLM may have limitations in transferring pretraining knowledge to downstream generalization.

Similar trends can be observed in BitNet b1.58, suggesting that this discrepancies between perplexity and downstream task performance may be a common limitation of low-bit or sparse model architectures. These results highlight the need for further development of scalable training strategies to achieve strong performance on both perplexity and downstream tasks while maintaining model efficiency.

## 5 Discussion

### 5.1 Advantages of TicketLLM over Quantized Models

In the previous section, we compared TicketLLM with BitNet b1.58 and dense baselines, and all models are trained from scratch on the same number of training tokens. While its training tokens are limited compared to state-of-the-art open-source LLMs, it has been observed that pre-training with low-bit representations from scratch can outperform quantized models where post-training quantization (PTQ) or quantization-aware training (QAT) methods are applied to open-source LLMs. These observations indicates that TicketLLM has the potential to achieve competitive performance with various quantization methods applied to open-source models, even with its current training limitations. This section explores this possibility by comparing TicketLLM with quantized models obtained by applying PTQ or QAT method to MobileLLM (Liu et al., 2024b), an open-source LLM designed for edge deployment. The model architectures of MobileLLM and

TicketLLM differ due to their distinct design objectives: MobileLLM modifies LLaMA architectures for edge deployment, while TicketLLM follows BitNet's architectural scaling approach. To enable fair comparison across different architectures, we focus on performance comparisons across models with closely matched parameter counts. All models in PTQ and QAT are quantized into 2-bit to align the storage of 2-bit TicketLLMs. All results of other quantization methods are cited from Liu et al. (2025). Table 3 summarizes the results. This table includes both PTQ methods such as RTN (Li et al., 2020), GPTQ (Frantar et al., 2022), AWQ (Lin et al., 2024), OmniQuant (Shao et al., 2023), and SpinQuant (Liu et al., 2024a), and QAT methods like LLM-QAT (Liu et al., 2017) and ParetoQ (Liu et al., 2025). Since previous methods did not evaluate on COPA, MMLU, and LLAMBDA, we omit these results from this table.

Compared with PTQ methods, Ada-Sup in TicketLLM demonstrates superior performance under 2-bit storage per parameter despite being trained with $2.4\times$ to $31\times$ fewer tokens than the base MobileLLM. These results reflect the inherent difficulty of applying PTQ in such extremely low-bit representations, where significant accuracy degradation is often observed. In contrast, TicketLLM, which involves parameter learning with low-bit representations take advantage of such constraints. These findings are consistent with the observations made in BitNet papers (Wang et al., 2023; Ma et al., 2024), emphasizing the importance of training under low-bit precision.

Compared with QAT approaches that allow parameter updating, TicketLLM achieves competitive performance on downstream tasks. Specifically, TicketLLM outperforms LLM-QAT by 1.5% to 9.4% in average downstream accuracy. Although ParetoQ, which utilizes sophisticated low-bit representation, demonstrates higher accuracy across all parameters, it sacrifices model sparsity due to the symmetric 2-bit quantization. This feature potentially limits its deployment efficiency, particularly on domain-specific hardware implementation for sparse low-bit models.

## 5.2 Further Research Directions

While our experimental results demonstrate the effectiveness of TicketLLM in achieving competitive performance with low-bit and sparse representation, several promising research directions remain to fully realize its potential. In this section, we outline two key research directions that could significantly advance the field: (1) algorithm-hardware co-design to exploit the efficiency benefits of TicketLLM, and (2) methods for effectively integrating knowledge from existing pre-trained models to reduce training costs while maintaining performance.

### Algorithm and Hardware Co-design for TicketLLMs

While GPUs remain the dominant platform in modern deep learning, their increasing energy demands pose critical sustainability challenges as LLM deployment scales. Relying exclusively on GPUs for all LLM workloads is not a viable long-term solution. Specialized hardware tailored to energy-efficient AI computing can significantly reduce energy requirements and operational costs. Thus, exploring such hardware is essential not only for accelerating inference but also for ensuring that future LLM systems remain scalable and sustainable.

Although specialized hardware addresses energy efficiency at the system level, complementary algorithmic approaches are equally critical for achieving favorable trade-offs between hardware efficiency and model performance. TicketLLM represents such an algorithmic direction, leveraging fixed random numbers, unstructured sparsity, and mixed-precision to improve the balance between hardware efficiency and model performance.

To realize the practical advantages of TicketLLM, hardware integration is essential for three key reasons. First, TicketLLM relies on an SLT-based structure, which leverages random weights for model compression. These features are known to be well-suited for specialized hardware integration to improve their efficiency (Hirose et al., 2022; Yan et al., 2025). Hardware support for random number generators enables random weights to be reconstructed on the fly, contributing to reduction of both capacity and bandwidth of weight memory. Specifically, to infer TicketLLM with effective weights equivalent to $n$-bit precision, the inference system only requires loading $(n-1)$-bit data, thereby reducing the required memory bandwidth. This 1-bit reduction substantially decreases energy consumption, as memory access requires approximately $640\times$ more energy than arithmetic operations (Horowitz, 2014). While random number generation incurs some computational

overhead, prior work has shown that lightweight generators such as Xorshift are sufficient to preserve model performance, making the computational overhead negligible.

Second, TicketLLMs exhibit unstructured sparsity, which is better suited for domain-specific hardware implementations than for general-purpose processors such as GPUs, TPUs, or CPUs (Parashar et al., 2017; Gondimalla et al., 2019). A key advantage of unstructured sparsity is its superior accuracy compared to structured sparsity at the same sparsity level (Frantar & Alistarh, 2023). If hardware efficiently supports unstructured sparsity, it would enable serving more accurate LLMs without compromising inference speed. Moreover, given the same target accuracy, unstructured sparsity enables the deployment of smaller models, further improving energy efficiency. Such unstructured sparsity can be exploited by some hardware mechanisms such as fine-grained scheduling and unnecessary operation skipping.

Third, the basic operations in TicketLLM are mixed-precision matrix multiplications, multiplying lower-precision weights with higher-precision activations. Such operation cannot be fully leveraged in current GPUs due to its reliance on dequantization-based approaches (Mo et al., 2025). In contrast, specialized hardware that natively supports mixed-precision operations can process $2\times$–$4\times$ more operations compared to existing GPU-based approaches (Mo et al., 2025). Through integration with custom hardware implementations, TicketLLMs can efficiently process mixed-precision computations without the overhead of dequantization, thereby achieving higher throughput and energy efficiency.

These aspects highlight the need for algorithm-hardware co-design, where hardware architectures directly exploit the model's characteristics to unlock the full efficiency of TicketLLM.

Another promising direction for hardware-software co-design is the exploration of semi-structured sparsity for TicketLLM. Semi-structured patterns, such as 2:4 sparsity, are supported by modern hardware (e.g., NVIDIA Ampere) and can improve execution efficiency without requiring custom hardware. Since TicketLLM achieves higher sparsity than BitNet b1.58, introducing such constraints is feasible without sacrificing model performance. This extension also enables future work on co-optimizing pruning strategies and hardware-aware model design, which can expand the applicability of TicketLLM.

**Integration of Knowledge from Pre-trained Models**

While the scaling performance of TicketLLMs in pre-training is promising, leveraging pre-training knowledge remains an important research direction in real-world scenarios. This importance is supported by Liu et al. (2025), which demonstrate that integrating fine-tuning with ternary representations can outperform BitNet b1.58, despite using only 33% as many tokens for fine-tuning as BitNet used for pretraining. Unlike quantization-based approaches, leveraging the pre-trained knowledge in SLT-based models is challenging due to random weights.

Distillation is a simple solution that leverages pre-trained features in SLT-based models. Hirose et al. (2022) propose a simple distillation approach that distills pre-trained weights into score distributions, resulting in improved performance. However, this approach still requires the same training as the original SLT-based models. Thus, incorporating pre-trained knowledge into SLT-based models to reduce training costs is still an important direction for future research.

## 6 Conclusion

This paper introduces Adaptive Supermasks (Ada-Sup), an efficient supermask optimization method that supports extremely low-bit sparse representations. Experimental results demonstrate that Ada-Sup offers superior trade-offs between training efficiency and model performance compared to existing SLT methods. Based on this method, we develop TicketLLM, a Transformer architecture that integrates Ada-Sup to achieve low-bit compression without training weights. Our evaluation shows that TicketLLM, powered by Ada-Sup, demonstrates superior scaling trends for dataset size and effective model size, particularly in the sub-10GB regime that is critical for edge device deployment. These findings underscore the potential of SLTs for enabling efficient low-bit representations, offering a promising solution for scalable model compression in LLMs through a concurrent blend of pruning, quantization, and random weights.

## Acknowledgment

This work was carried out using the TSUBAME4.0 supercomputer at Institute of Science Tokyo. This work was supported in part by JSPS KAKENHI Grant Numbers JP23H05489, JP25K03092, and JP23KJ0955, and by JST-ALCA-Next Japan Grant # JPMJAN24F3.

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

Table 4: Model configuration. We scale the number of layers and model dimensions while head dimension keeps constant.

| $N$ | $n_{\text{layers}}$ | $n_{\text{heads}}$ | $d_{\text{model}}$ | $d_{\text{head}}$ | $d_{\text{FFN}}$ |
|------|------|------|------|------|------|
| 0.1B | 12 | 12 | 768 | 64 | 2,048 |
| 0.3B | 24 | 16 | 1,024 | 64 | 2,731 |
| 0.7B | 24 | 24 | 1,536 | 64 | 4,096 |
| 1.3B | 24 | 32 | 2,048 | 64 | 5,460 |

Table 5: Hyperparameters for training. Batch size is denoted as BS, while learning rate is described as LR. Warmup and Steps describe their respective number of iteration. Tokens means the number of training tokens.

| Model | TPP | LR | BS | #Warmup | #Steps | #Tokens |
|------|------|------|------|------|------|------|
| 0.1B | 320 | 6.6e-4 | 512 | 334 | 33,438 | 32B |
| 0.3B | 320 | 5.0e-4 | 512 | 1,022 | 102,216 | 56B |
| 0.7B | 320 | 3.9e-4 | 512 | 2,224 | 222,434 | 223B |
| 1.3B | 320 | 3.1e-4 | 512 | 3,886 | 388,680 | 408B |

## A  Training configurations

We provide all model configurations in Table 4 and hyperparameters in Table 5.

## B  Related Works

### B.1  Strong Lottery Ticket

Strong Lottery Tickets (SLTs), which are accurate subnetworks within random weights, demonstrated additional compression opportunities that leverage random weights. After Zhou et al. (2019) have established the basic concept of SLTs, following research focuses on algorithm improvement to find better subnetworks. Ramanujan et al. (2020) introduce the EdgePopup algorithm that selects top-k% of score parameters, making it possible for SLTs to find good subnetwork on practical datasets. Zhou et al. (2021) improve SLT accuracy in highly sparse regions by introducing ProbMask. Koster et al. (2022) introduce FixedThreshold, which improves both training efficiency and performance in highly sparse regions. Moreover, other research focuses on improving SLT model capacity by expanding its concept to multiple random seeds (Chijiwa et al., 2021) or multiple supermask (Okoshi et al., 2022), improving model performance, especially on large-scale datasets. Despite the potential of SLTs, it is difficult to balance training efficiency and scalability, which are crucial for large-scale pre-training in large language models. Instead, our proposal can balance training efficiency and scalability by leveraging a quantization-based approach to score parameters.

Another line of research focuses on expanding its application. García-Arias et al. (2023) integrate SLT with network folding, demonstrating the existence of SLT in folded neural networks. It also shows that network folding can help identify smaller SLTs for ResNets (He et al., 2016; Zagoruyko & Komodakis, 2016). Huang et al. (2022) extend SLTs to graph neural networks, claiming that SLTs not only match the performance of trained models but also alleviate the over-smoothing problem, a major challenge in scaling deeper GNNs (Cai & Wang, 2020). The following studies focuses on improving the algorithm (Yan et al., 2024) or expanding GNN architectures (Ito et al., 2025). Shen et al. (2021) are the first to apply SLTs to Transformers. Exploring single-layer encoder-decoder Transformers reveals that SLTs are also effective in machine translation. Despite these advancements, a major limitation of existing research is its primary focus on shallow Transformers, which differ significantly from deeper, decoder-only architectures of large language models (LLMs). Our work addresses this limitation by extending SLTs to LLMs through Ada-Sup. This novel approach overcomes the challenges of applying SLTs to large-scale pre-training, bridging the gap between prior research and modern neural networks.

### B.2  Quantization and Pruning

Quantization and pruning are commonly used as model compression methods. Quantization reduces both computational costs and storage by reducing the precision of parameters. While some research focuses on quantizing both weights and activations, weight-only quantization remains more actively explored in LLMs, as increasing parameters are critical for morel serving. Weight-only quantization can be categorized into post-training quantization (PTQ) and quantization-aware training (QAT), depending on whether fine-tuning is performed after quantization. PTQ has gained significant attention due to its simplicity and efficiency, and several research such as GPTQ (Frantar et al., 2022) and AWQ (Lin et al., 2024) have demonstrated strong performance on 3-bits or 4-bits quantizations. Recent studies have focused on 2-bit quantization (Shao et al.,

2023), accepting marginal accuracy degradation as a trade-off. Nevertheless, QAT (Liu et al., 2023; Wang et al., 2023; Liu et al., 2025) has garnered increasing attention for its ability to support extremely low-bit quantization (e.g., below 2 bits), which is often difficult to achieve with PTQ alone. Unlike conventional quantization methods that apply PTQ or QAT to pre-trained LLMs, BitNet (Wang et al., 2023) proposes a fundamentally different approach by training transformer models from scratch using extremely low-bit (e.g., 1-bit) weights, offering a novel perspective on low-bit LLMs and scaling behavior.

Pruning improves model efficiency by removing unnecessary weights. Pruning is generally categorized into unstructured and structured pruning, depending on the granularity of the introduced sparsity. Due to the efficiency in general-purpose hardware, semi-structured sparsity (Frantar & Alistarh, 2023; Sun et al., 2023; Fang et al., 2024) is mainly explored in LLMs, even though unstructured sparsity can maximize their efficiency by integrating custom hardware implementation.

These two compression techniques are not mutually exclusive and can be easily combined in practice. Han et al. (2016) is a pioneered work combining pruning and quantization, demonstrating there multiple compression techniques can be easily integrated. Ma et al. (2024) extend the numerical representation of BitNet by shifting from binary to ternary weights. This research has successfully achieved extremely low-bit representations whose performance is comparable with dense models by integrating pre-training with low-bit representations. SLTs follow a similar procedure of concurrent pruning and quantization but also leverage random weights for additional storage savings.

