# OpenReview forum: "TicketLLM: Next-Generation Sparse and Low-bit Transformers with Supermask-based Method"
_TMLR — Accepted by TMLR_

### Review · Reviewer_hbZm · 2025-07-21

**Summary Of Contributions:**

**Summary**: the authors introduce TicketLLM as a new paradigm for edge-optimized LLMs through their new training method **Ada-Sup** (adaptive supermask) which is a quantization-based method for efficiently learning supermasks over fixed random weights $w_{ij} \in \{-c, +c\}$. This cuts down supermask training time compared to prior high-performant SLT methods. They scale their TicketLLM up to 1.3B parameters and provide scaling studies from 110M-1.3B parameters. They achieve 50% sparsity with 2 bits per connection and match/beat low-bit baselines like BitNet on perplexity and downstream tasks.

**Strengths**
- First work to show that SLT can be made to scale in ~1B parameter transformer language models, opening up for new work on the scalability of SLT.
- Elegant and simple mask learning with simple quantization operations, and competitive performance against BitNet and other post-training quantization methods.
- Shows good scaling to larger models with more parameters (Figure 6b) and might close to gap to 16-bit transformers.
- Shows better scaling than BitNet in utilizing more tokens per parameter (TPP). Whereas BitNet plateaus ~160 TPP, TicketLLM continues to improve on evaluation loss (though not as clearly on downstream tasks).

**Weaknesses**
- The proposed method yields diminishing returns for higher-bit representations. While the binary supermask yields up to a 2x memory reduction over ternary weights (which may be stored in 2 bits), using an $n$-bit multimask to represent an $(n+1)$-bit symmetric quantized representation yields only an $(n+1)/n$ reduction in memory.
- Equation 6 in Section 3.1 shows that $\gamma$ is computed as the mean of absolute values of the parameters. Unless the distribution of scores is uniform (which I consider unlikely, but difficult to judge without knowing the initialization scheme), the quantization in Equation 5 and 6 is sub-optimal as it will use equally sized bins for the all score values in the range $[0, 2^n-1]$.
- Figure 4: it is hard to make deductions from their plot since they do not provide an isoFLOP-comparison curve.
- The parameter scaling study uses a fixed TPP of 320 which is sub-optimal for BitNet and may therefore not quite be a fair comparison.
- No hardware acceleration results. The method relies on on-the-fly random-weight generation and unstructured sparsity, neither of which current general-purpose hardware like GPUs and TPUs handles well (to the best of my knowledge). It is claimed that 2:4 structured sparsity could be leveraged with 50% unstructured sparsity, but no results are provided (and there will likely be a performance drop for 2:4 structured sparsity conversion).
- Mixed results in Figure 5b on downstream tasks. It's not clear how well Ada-Sup will scale to larger models trained on more tokens in terms of downstream task performance.
- This method required training LLMs from scratch, which is far more expensive than the post-training quantization methods that the authors compare against at the end of the paper.

**Additional Comments:**

- I would be curious how the execution time was measured for Figure 2. Are these just naive PyTorch implementations, or optimized CUDA kernels?
- The scaling laws in Figure 6 predict that BitNet will outperform TicketLLM at larger scale. Do the authors have any suggestions for why this might be the case, and what can be done to close the gap? The sentence "*improved trade-offs between effective model size and loss performance take advantage of deployment on edge devices where low-bit representations are crucial*" that is given as an explanation on page 9 is not comprehensible to me.
- Can this pattern of quantization and sparsity be accelerated on hardware as presented in [Pierro et al, 2025](https://openreview.net/forum?id=UNrfYfbLZ3) and if so/not, what are additional challenges for hardware acceleration of TicketLLM?

**Audience:**

Yes

**Audience Explanation:**

This is an interesting approach to quantization and pruning, with demonstrated scalability to 1B parameter LLMs. The community will benefit from the lessons presented in this paper and the problems that are highlighted herein, such as the limited performance on downstream tasks. Moreover, this can be a promising direction for computer architecture and hardware design/optimization for low-bit and sparse LLMs.

**Broader Impact Concerns:**

No concerns.

**Claims And Evidence:**

Yes

**Claims Explanation:**

The scalability and performance of their newly proposed method is well-documented in their experiments and their report.

**Requested Changes:**

- Discussion of score distribution. Concretely, please specify how the scores are initialized for training. See weakness point #2.
- Figure 6a uses the "effective model size" on the x-axis, but this is just counting the number of non-zero parameters without considering the sparse encoding scheme. In practice, the model size for sparse models is unattainable and it is more realistic to provide the model size when using sparse encoding schemes (e.g., COO, CSR, CRS, CSC, CCS). Alternative, the authors may use "effective parameter count" with a discussion about the limitations to illustrate idealized memory savings.

Minor changes:
- instead of giving an absolute perplexity improvement, a relative improvement might be more interpretable in the abstract.
- type in section 2.1: gambel-softmax → gumbel softmax
- Section 2.3: "multiple pre-training is required for one model" → "multiple pre-training runs are required for one model"
- The center part of Figure 3 is not easy to parse, it would be very beneficial for the reader if this could be illustrated better. Also, what method is presented on the left? Is that just the standard SLT for a binary supermask, or something else? A reference might be helpful in the caption. And the "quant. range" on the right is more of a "quant. scale", isn't it?
- Section 3.3: "We follow the LLaMA to design the Transformer, (...)" → "We follow the LLaMA architecture, (...)"
- Section 4.2: "We also include reproduced LLaMA models as a baseline." → "We also include reproduced LLaMA model results as a baseline."
- If possible, it would be helpful to repeat the main assumptions of Gadre et al (2024) in the main text for the scaling law in Eq. 7.
- In Table 3, FP results may be highlighted as such to make parsing the table easier.

---

> ### Author Response · Authors · 2025-08-14
> **Response to Reviewer hbZm**
>
> Thank you for your thoughtful and constructive feedback. We greatly appreciate your time and effort in reviewing our manuscript. Below, we respond to each reviewer’s specific concerns in turn.
>
> > Diminishing returns for higher-bit representations.
>
> Thank you for raising the important consideration. As the reviewer correctly pointed out, the memory reduction ratio achieved by our proposal decreases as the bit-width increases. However, our method focuses on improving trade-off between model size and performance on language tasks in extremely low-bit representation such as 1 bit or 2 bits. In such cases, the reduction ratio is substantial. For example, in the 2 bits case, our method achieves approximately a 33% memory saving to represent a 3 bits symmetric distribution.
>
> Moreover, our approach induces higher sparsity, which can lead to additional memory reduction and computational efficiency. Therefore, while the relative memory benefit naturally diminishes as bit-width increases, our method remains effective and impactful in low-bit representations that are particularly relevant for deployment in resource-constrained environments.
>
> > Suboptimal quantization
>
> Thank you for the insightful comment. As the reviewer correctly pointed out, non-uniform quantization is not optimal for minimizing quantization error when the underlying distribution is non-uniform. Indeed, models using relatively higher bit-widths, such as 4 bits or 8 bits, often achieve better performance with floating point (non-uniform) formats compared to integer (uniform) quantization.
>
> However, extremely low-bit representations (e.g., 1 bit or 2 bits) do not have sufficient bit-width to support floating-point representations. To address this, some approaches use codebook-based quantization methods [1, 2], but their performance remains sub-optimal. For example, PV-Tuning [2] reports that similar 1B models show an average performance drop of 2.37% across five tasks compared to our proposed method. Although these results are not reported in the original PV-Tuning paper, we obtained them from its implementation on Hugging Face [3]. For this reason, we do not include them in our main paper. Nevertheless, the observed accuracy gap supports our point that minimizing quantization error is not always the most critical factor in extremely low-bit scenarios.
>
> |Base Model|#-bits|ARCe|ARCc|HS|PIQA|Wge|Avg.|
> |---|---|---|---|---|---|---|---|
> |PV-tuning LLaMA 3.2 1.23B|2|58.96|27.13|40.34|70.67|55.64|50.55|
> |Ours TicketLLM 1.3B|2|54.29|33.02|52.03|70.35|54.93|52.92|
>
> > Iso-FLOP-comparison in Figure 4
>
> Thank you for the insightful comment. To address this point, we have revised the x-axis of Figure 4 to show the effective model size, defined as #bits x (1-sparsity), which provides a more meaningful iso-comparison across different bit-widths.
>
> > Unfair TPP selection for parameter scaling
>
> We appreciate the reviewer’s concern regarding the choice of a fixed TPP of 320. However, we believe this setting aligns well with recent trends in LLM pre-training and provides a fair basis for comparison. Although our experiments are conducted on smaller-scale models, observing scaling behavior under higher TPP settings allows us to estimate performance trends of practical large-scale pre-training.
>
> Recent LLMs have generally been trained with higher TPP values to achieve better performance with lower inference cost. From this perspective, models that achieve better performance under higher TPP are particularly desirable.
>
> Our selection of TPP=320 reflects this trend, aiming to evaluate models in realistic pre-training scenarios. In addition, we observe that BitNet also benefits from increasing TPP in the larger parameters, as shown in the following table, indicating that 320 TPP is not necessarily a sub-optimal setting for BitNet.  Importantly, this observation remains consistent with the results in Figure 5 since the performance improvement observed in TicketLLMs is larger than that of BitNet under the same TPP increase.
> |Model|#Bit|Param|PPL (TPP=160)|PPL (TPP=320)|
> |---|---|---|---|---|
> |BitNet|2|700M|**16.46**|15.35|
> |TicketLLM|2|700M|16.51|**15.06**|
> |BitNet|2|1.3B|**14.57**|13.62|
> |TicketLLM|2|1.3B|14.61|**13.54**|
>
> Due to limitations in available compute and dataset size, we were unable to conduct experiments at even higher TPPs for parameter scaling. Nevertheless, we believe that the results at TPP = 320 already provide meaningful insight into the effectiveness of the proposed method, particularly in contexts aligned with modern LLM development.
>
> [1] Tseng, A. et al. QuIP#: Even Better LLM Quantization with Hadamard Incoherence and Lattice Codebooks.
>
> [2] Malinovskii, V. et al. PV-Tuning: Beyond Straight-Through Estimation for Extreme LLM Compression
>
> [3] ISTA DASLab. https://huggingface.co/ISTA-DASLab/Llama-3.2-1B-AQLM-PV-2Bit-2x8

---

> ### Author Response · Authors · 2025-08-14
> **Response to Reviewer hbZm**
>
> > No hardware acceleration results
>
> We appreciate the reviewer’s thoughtful observation. We agree that our current work does not include hardware acceleration results, and we acknowledge our algorithm features, such as on-the-fly randomness, unstructured sparsity, and mixed-precision operations, are not well supported by existing general-purpose hardware such as GPUs and TPUs.
>
> As discusses in Section 5.2, we believe that some domain-specific hardware architectures are better suited for our proposed method. For example LUT-Tensor Core [4] can process approximately twice as many operations as an H100 GPU by supporting mixed-precision matrix multiplication. In addition, while on-the-fly random weight generation is not yet widely supported by general purpose hardware, some approaches [5, 6] have demonstrated promising directions for leveraging such randomness.
>
> These findings highlight the importance of designing efficient algorithms in conjunction with domain-specific hardware. We believe our work takes a step toward exploring algorithmic possibilities that can inspire future co-design of models and hardware.
>
> We also hope that our work motivates further research into energy-efficient hardware architectures capable of supporting unstructured sparsity and mixed-precision operations, as these features are increasingly critical for the deployment of low-bit LLMs in real-world scenarios. We will incorporate this discussion into the main manuscript.
>
> > No clarification for semi-structured sparsity
>
> Thank you for your insightful comment, We agree that our paper does not provide experimental results for 2:4 semi-structured sparsity, and that introducing semi-structured restriction may lead to a performance drop in some cases.
>
> However, as noted in Section 5.2, our intention was to present this as a promising direction for future algorithm and hardware co-design, rather than a claim of current implementation. Given that TicketLLM achieves relatively high sparsity with unstructured one, we believe it is feasible to impose additional structure with less degrading model performance compared with existing approaches though it might be require careful pruning strategy.
>
> Future work could explore training algorithms that directly incorporate semi-structured sparsity, potentially improving hardware efficiency while maintaining model quality.
>
> > Mixed downstream performance
>
> We appreciate the reviewer’s observation regarding the mixed downstream performance shown in Figure 5b. As the reviewer pointed out, the downstream performance of TicketLLM does not scales as increasing training datasets.
>
> However, we would like to note that this limitation is not unique to TicketLLM. Similar trends can be observed in BitNet, indicating that this may be a broader challenge inherent to the extremely low-bit model rather than a shortcoming specific to our method.
>
> Addressing this issue is an important open problem. We believe that future work on low-bit optimization techniques will be critical to improving generalization in such models.
>
> > Cost of from-scratch training
>
> We appreciate the reviewer’s point and agree that our method requires training LLMs from scratch. However, the primary goal of our method is to enable highly accurate LLMs in extremely low-bit representations (e.g., 1 or 2 bits), where existing PTQ techniques often suffer from substantial performance degradation. In such cases, we believe that incorporating quantization into the pre-training process can offer a more effective approach toward building practical low-bit LLMs.
>
> In addition, compared to QAT-based approaches, our method achieves promising results, suggesting that large-scale pre-training with a comparable number of tokens to existing LLMs can lead to further improvements. These results highlight that pre-training with a carefully designed low-bit numerical representation offers a promising direction for developing efficient and scalable LLMs.
>
> We also acknowledge that reducing training costs remains an important challenge. As discussed in Section 5.2, one promising future direction is to combine supermask-based methods with pre-training knowledge to enable more efficient parameter tuning within low-bit models.
>
> [4] Mo, Z. et al. LUT Tensor Core: A Software-Hardware Co-Design for LUT-Based Low-Bit LLM Inference.
>
> [5] Hirose, K. et al. Hiddenite: 4K-PE Hidden Network Inference 4D-Tensor Engine Exploiting On-Chip Model Construction Achieving 34.8-to-16.0 TOPS/W for CIFAR-100 and ImageNet.
>
> [6] Yan, J. et al. BingoGCN: Towards Scalable and Efficient GNN Acceleration with Fine-Grained Partitioning and SLT.

---

> ### Author Response · Authors · 2025-08-14
> **Response to Reviewer hbZm**
>
> > Discussion of score distribution
>
> Thank you for pointing out. The scores are initialized using a normal distribution with a standard deviation of 0.02, following the initialization scheme used in LLaMA 2 weights.
>
> We will updated Section 4.2 (Experimental setup) to include this detail for clarity.
>
>
> > Choice of effective model size
>
> We appreciate the reviewer’s thoughtful comment. As correctly pointed out, the “effective model size” used in Figure 6 reflects an idealized count of non-zero parameters and does not account for the overhead of sparse encoding schemes. We agree that in practical implementations, such overhead can be non-negligible and optimal encoding may vary depending on both hardware architecture and target sparsity.
>
> Our motivation for using this effective model size to enable a hardware-agnostic comparison that isolates the algorithmic effects of efficiency. Thus, we chose to report the effective model size as a consistent baseline across models.
>
> However, we acknowledge that the term "effective model size" may have been misleading. To improve clarity, we have revised the terminology in the updated manuscript and use "effective bit count" instead.
>
> We agree that this is a relevant limitation, and we will explicitly clarify this point in the revised manuscript.
>
> > Scaling law assumption
>
> Thank you for the helpful suggestion. We agree that explicitly stating the key assumptions for the scaling estimation would improve clarity. In the revised version, we will update the main text to include the assumptions underlying the simplified scaling law in Equation 7. Specifically, we clarify that the original formulation assumes
>
> $L(N, D) = E + AN^{-\alpha}+BD^{-\beta}$,
> and that the simplification to
>
> $L(N) = E + A'N^{-\alpha}$,
>
> is based on two assumption (1) $\alpha = \beta$, and (2) a fixed TPP $D = 320N$, leading to $A' = A + 320B$. We will add this deriviation before Equation 7 to improve clarity.
>
> > Measurement way of execution time in Figure 2
>
> Thank you for the question. The execution time results shown in Figure 2 were obtained using naive PyTorch implementations, without any custom kernel optimizations or fused CUDA operations. We will update the caption of Figure 2 to clarify this implementation detail.
>
> > Prediction of BitNet outperforming
>
> Thank you for the insightful question. We currently do not have a definitive exploration for difference in parameter scaling between TicketLLM and BitNet, but we suspect the discrepancy may be influenced by sub-optimal hyperparameter settings, as low-bit models are often more sensitive to such choices.   Further investigation is needed to fully understand the behavior.
>
> However, we would like to highlight that TicketLLM demonstrates better scaling with respect to effective model size, achieving lower loss per effective parameter. This suggests that TicketLLM provides more feasible solution in resource constraint environment.
>
> To further improve TicketLLM’s scalability, we believe that enhancements in training setup and data quality may help close the performance gap at larger model scales.
>
> As for the sentence on page 9, we agree that it was unclear. We will revise it as follows:
>
> *better trade-offs between effective model size and performance demonstrates TicketLLMs provide a suitable option in resource constraint devices.*
>
> > Hardware acceleration feasibility
>
> Thank you for your insightful question. As shown in [7], there is promising evidence that low-bit sparse LLMs can be efficiently accelerated on the Loihi 2 chip. Additionally, according to [8], this hardware also supports random bit generation. Given these features, Loihi 2 may be capable of processing TicketLLMs with the hardware platform.
>
> However, architectural limitations in Loihi 2 may lead to sub-optimal performance in TicketLLMs. For example, the need to convert artificial neural network to sigma-delta neural network [9] restricts the numerical format for quantization scales in TicketLLMs, potentially leading performance degradation. In addition, Loihi 2 currently supports only 8 bits weights, which may result in inefficient memory usage in extremely low-bit models. Furthermore, its limited supporting of activation functions requires modifications to the model architecture of TicketLLMs. These limitations suggest that further hardware investigation is required to fully accelerate TicketLLMs.
>
> We will include the discussion of Loihi 2 in Section 5.2, as its architecture is highly relevant to the further research direction.
>
> > Abstract modification, typo, improving wording, figure 3 clarity, and improving tables
>
> Thank you for pointing this out. We will address these points in the revised version.
>
> [7] Abreu, S. et al. Neuromorphic Principles for Efficient Large Language Models on Intel Loihi
>
> [8] Orchard, G. et al. Efficient Neuromorphic Signal Processing with Loihi 2.
>
> [9] Brehove, M. et al. Sigma-Delta Neural Network Conversion on Loihi 2.

---

> > ### Author Response · Authors · 2025-10-13
> > **Response to Reviewer hbZm**
> >
> > Furthermore, in response to the feedback, we have revised the manuscript accordingly. All modifications are highlighted in red in the revised version. Below, we summarize the key updates made specifically in response to the reviewer's comments.
> >
> > > Add the score initialization
> >
> > We have added the description of score initialization in the *Experimental Setup* section:
> >
> > - “Scores are initialized using a normal distribution with a standard deviation of 0.02.”
> >
> > > Improving the clarity of model size computation
> >
> > To tackle the limitations of model size computation, we have revised manuscript as follows:
> >
> > - Replaced “effective model size” with “effective bit count.”
> > - Added a note on the limitation of this metric:
> > ”While this bit count doesn't reflect the actual model size due to the lack of considering the overhead of sparse encoding, it serves as a useful metric for comparing the theoretical storage requirements across different quantization schemes.”
> >
> > > Update Figure 3
> >
> > To improve readability, we have updated Figure 3.
> >
> > - Reorganized the layout in the middle part to make it visually clearer
> > - Updated the label on the left side from *“quant. range”* to *“quant. scale”*
> > - Added the phrase *“including both single and multivalued mask”* to the caption
> >
> > > Strengthen the justification for exploring the co-design of hardware and algorithms
> >
> > We have revised Section 5.2 to provide a more comprehensive justification for hardware-algorithm co-design. The revisions include:
> >
> > - A detailed analysis of GPU energy consumption challenges and the critical need for specialized hardware
> > - An examination of how random weights reduce memory bandwidth requirements
> > - A comparative evaluation demonstrating the advantages of unstructured sparsity over structured approaches
> > - A quantitative assessment highlighting the benefits of native mixed-precision hardware support
> >
> > > Implementation details for Figure 2
> >
> > We have added a description of the implementation framework to Figure 2's caption.
> >
> > - Execution time is measured using the native PyTorch implementation (without custom CUDA kernels or third-party optimizations).
> >
> > > Repeating the main assumptions in scaling law
> >
> > We have revisited the assumptions for Gadre et al. with the original scaling law formulation in Section 4.3.
> >
> > > Other improvement
> >
> > We have improved the wording, enhanced table highlighting, and fixed typos.
> >
> > We are grateful to the reviewer's thoughtful and detailed feedback, which helped us improve the manuscript substantially.

---

### Review · Reviewer_2H8V · 2025-08-19

**Summary Of Contributions:**

**Summary**

This paper proposed a multi-value Strong Lottery Ticket (SLT) method using a quantization scheme instead of top-k or fixed thresholding methods during supermask generation. SLT in general tries to compress the model weights with a low-bit mask multiplied with a randomly generated weight tensors, such as (W_random * supermask_Nbit). Since the weight is randomly generated, only the low-bit mask needs to be stored, therefore, the compression is achieved. The mask is derived from a trained score tensor (which is always non-negative in this case) by scaling its mean to (2^n-1) and quantize to integers. The impact on perplexity and downstream accuracy by model size and amount of data used in training are also investigated. Compared to ternary BitNet b1.58, the proposed method showed comparable performance for model size ~0.7b to 1.3b. When compared with other popular quantization methods, e.g. SpinQuant, using 2bits settings, the proposed method outperformed most of the options except ParetoQ. The critical issues needed to be addressed were also discussed in Section 5.2.

**Strength**

1. Good amount of experimental data. Many experiments clearly require tens of GPU hours or more to complete.
2. Easy to follow. Reasonable improvements upon other SLT methods.

**Weakness**

As pointed out by the author in Section 5.2, the two main concerns/limitations of this work are: 1) requires a few hardware features that do not exist in modern GPUs, mainly efficient, parallelized random number generator and unstructured sparsity support, and 2) requires training from scratch for each model which incurs considerable overhead. These would be considered very high adoption barriers, however, the corresponding justifications do not seem to be sufficient. For example,

1. **Estimate of potential benefit in inference**.  Since the proposed random weights is {-1, +1}, one may consider (W_rand * mask_nbits) equivalent to nbits quantized weight, (which could eliminate the need for specialized random number generator.) Under this assumption, the inference speed would be equivalent to corresponding quantized model. Taken the sparsity into account and assuming the achievable speed-up by unstructured sparsity can be close to that provided by GPU's structured sparsity, it's unclear how much additional gain in inference speed can be achieved compared to typical modern GPU. The author may want to elaborate a bit more to justify the investment in new HW and additional training for each model.

2. **Based on relatively old model Llama2** All the experimental data in this work are based on Llama2, however, Llama3 has more optimized features, e.g. GQA. It would be nice to include at least some Llama3 data, unless there exists incompatibility of the method with newer Llama architecture.

3. **Quantization of score tensor** For binary case, Eq. 4 maps `abs.mean` to 1, which means 50% of the number (value > mean) will be clamp to 1 and 50% of the elements will be <1. However, due to the rounding operation, only elements with values 0 < val < 0.5 will become 0, which doesn't make up 50% sparsity (i.e. 50% of the tensor elements should be 0.) Similarly, for N-bit cases using Eq. 5, the resulting sparsity seems to be lower than 50% as well. Could author clarify the 50% sparsity computation of using Eq.4/Eq 5?

**Audience:**

Yes

**Audience Explanation:**

Low-bit compression method in general is of great interests, Lottery Tickets and supermask method might not be the most effective one, but could serve as a reference for other quantization methods.

**Broader Impact Concerns:**

No ethical concerns.

**Claims And Evidence:**

Yes

**Claims Explanation:**

Claim 1 -> Fig. 2 and 4.
Claim 3 -> Table 1, comparable results.

**Requested Changes:**

please see Weakness above.

---

> ### Author Response · Authors · 2025-09-11
> **Response to Reviewer 2H8V**
>
> Thank you for the thoughtful comments and constructive feedback. We greatly appreciate your time and effort in reviewing our manuscript. We would like to answer your comments as follows:
> > Justification for the investment in new HW
>
> Thank you for the valuable comments. We acknowledge the concern that introducing specialized hardware causes high adoption barriers. We share the same understanding that GPU is the governing platform in modern deep learning technology. However, from the perspective of power consumption, relying exclusively on GPUs poses a critical challenge. As LLM deployment scales, the energy demand of GPUs is becoming increasingly unsustainable. While GPUs are highly versatile, executing every LLM on GPUs is not a viable long-term solution. A new hardware that is tailored to energy-efficient computing of modern AI workloads can significantly reduce energy requirements, alleviating operational costs. In this sense, exploring and developing such a new hardware is not only desirable in accelerating inference speed but also necessary to ensure that future LLM systems remain both scalable and energy-efficient. Beyond the general need for specialized hardware, we believe exploring specific algorithms that can influence the trade-offs between hardware efficiency and model performance is crucial. In this context, TicketLLMs illustrates an algorithmic direction that leverages fixed random numbers, unstructured sparsity, and mixed-precision, enabling further improvements in balancing hardware efficiency and model performance.
>
> We acknowledge that our manuscript lacks sufficient justification for the investment in new hardware. Therefore, we will add a discussion of the energy challenges associated with GPUs in Section 5.2. Thank you very much for your indispensable comment, again.
> > Potential benefit of random weights in inference
>
> Random number generator reduces the memory access for weights. To infer TicketLLM with effective weights equivalent to n-bit, the inference system only requires loading (n-1)-bit data, reducing energy consumption for memory access that needs 640x higher energy than simple arithmetic [1]. This 1-bit reduction is substantial, especially in an extremely low-bit situation. Though the random number generator requires some arithmetic costs, prior work [2] has shown that simple generators such as Xorshift are sufficient to preserve model performance, making the overhead very small.
> > Potential benefit of unstructured sparsity
>
> The main motivation for adopting unstructured sparsity over structured sparsity is its potential to preserve model accuracy. As demonstrated in [3], unstructured sparsity generally delivers better accuracy than structured sparsity at the same sparsity level. Therefore, if hardware support for unstructured sparsity can provide comparable speed to that achieved on current GPUs, it would enable serving more accurate LLMs without compromising inference speed. In addition, given the same accuracy, unstructured sparsity enables the deployment of smaller models, facilitating energy-efficient inference systems if new hardware effectively supports unstructured sparsity. We believe these in both accuracy and energy efficiency motivate to explore investment in new hardware.
> > Additional gain in inference speed compared to typical modern GPU
>
> Supporting native mixed-precision operations can provide inference speedup compared with existing GPU implementations. As demonstrated in LUT-Tensor Core [4], implementations that efficiently combine low-bit weights with high-bit activations can process 2×–4× more operations over existing GPU-based approaches. Consequently, integrating mixed-precision into specialized hardware would allow future systems to serve larger and more accurate LLMs while further enhancing the inference speed.
>
> In summary, supporting these features with hardware implementation can simultaneously improve inference speed and energy efficiency, enabling the deployment of more scalable and sustainable LLM systems. Based on the above discussion, we will revise Section 5.2.1 as follows:
> - We will add a discussion at the beginning of Section 5.2.1 on the necessity on hardware beyond GPUs
> - We will include a discussion on the impact of reduced memory access using random weights on energy consumption
> - We will add a discussion on improving trade-off between accuracy and parameter count through the use of unstructured sparsity
> - We will provide quantitative results comparing the number of operations achieved by LUT-Tensor Core with mixed-precision support against GPUs

---

> ### Author Response · Authors · 2025-09-11
> **Response to Reviewer 2H8V**
>
> > Requiring training from scratch for each model
>
> We acknowledge that pre-training from scratch incurs considerable overhead. However, we believe the benefits of low-bit pre-training surpass the overhead it incurs. As illustrated by BitNet [5, 6], designing low-bit LLMs from scratch enables models to achieve remarkable performance in both pre-trained LLMs and instruction-tuned LLMs, in some cases reaching accuracy comparable to models trained with higher-precision formats such as FP16.
> As low-bit models are anticipated to deliver higher inference efficiency, pursuing these algorithmic directions is well motivated. From this point of view, this paper investigates the feasibility of new hardware-oriented LLMs that integrate quantization, unstructured pruning, and random weights.
>
> At the same time, we acknowledge that developing methods to reduce training costs will be a critical future research direction. For instance, distilling the knowledge from a larger TicketLLM into a smaller TicketLLM is a promising approach for reducing training costs for each model. As demonstrated in state-of-the-art language models [7], knowledge distillation for smaller models can eliminate the repetitive training effort, thereby reducing overall training costs. In this work, however, we focus on constructing the foundational TicketLLM itself along with revealing the nature of its scaling behavior, leaving such cost-reduction techniques for future work. To improve clarity, we will revise Section 5.2.2 as follows:
> - In Section 5.2.2, we will add a discussion of model distillation in relation to reducing training costs for smaller models.
>
> > Based on the relatively old model Llama2
>
> Thank you for the valuable comment. In this work, our primary focus was to capture general trends such as the validity of scaling laws by training and analyzing multiple models. To achieve this goal,  we selected the LLaMA2 architecture, which has already been used in prior scaling law analyses of low-bit LLMs [8]. We acknowledge that LLaMA2 is not the most recent architecture. However, for the purpose of investigating scaling laws, this architecture provides a well-established and suitable reference model. As noted in [9], scaling laws are considered relatively robust to architectural changes. Thus, while features such as GQA may influence absolute model performance, we believe they do not affect the existence of scaling laws themselves, which we believe is the main contribution of this manuscript. While our analysis does not exhaustively cover variations in model architecture, we believe the current contributions of TicketLLM provide sufficient and meaningful insights. At the same time, we fully agree that architectural improvements such as GQA are important for enhancing inference efficiency, and we plan to investigate in more depth in future work how the behavior of our method varies with different model architectures.
>
> > Quantization of score tensor
>
> Thank you for the insightful comment. We would like to clarify that the “50% sparsity” mentioned in the paper does not come directly from the mathematical form of the equations, but rather refers to the empirically observed sparsity in our experiments. As observed in Table 2 in the manuscript, our model achieves approximately 50% sparsity at any parameters within the range of our experiments. This sparsity does not vary significantly with the number of parameters, so we expect the same tendency to hold for other parameter scales as well. To improve clarity, we replace 50% sparsity in the abstract with around 50% sparsity.
>
> ---
> References:
>
> [1] Han, S. et al., Learning both Weights and Connections for Efficient Neural Networks
>
> [2] Hirose, K. et al., Hiddenite: 4K-PE Hidden Network Inference 4D-Tensor Engine Exploiting On-Chip Model Construction Achieving 34.8-to-16.0 TOPS/W for CIFAR-100 and ImageNet
>
> [3] Frantar, E. and Alistarh, D., SparseGPT: Massive Language Models Can Be Accurately Pruned in One-Shot
>
> [4] Mo, Z. et al., LUT Tensor Core: A Software-Hardware Co-Design for LUT-Based Low-Bit LLM Inference
>
> [5] Ma, S. et al., The Era of 1-bit LLMs: All Large Language Models are in 1.58 Bits
>
> [6] Ma, S. et al., BitNet b1.58 2B4T Technical Report
>
> [7] Qwen Team, Qwen3 Technical Report
>
> [8] Wang, H. et al., BitNet: Scaling 1-bit Transformers for Large Language Models
>
> [9] Keplan, J. et al., Scaling Laws for Neural Language Models

---

> > ### Author Response · Authors · 2025-10-13
> > **Response to Reviewer 2H8V**
> >
> > Furthermore, in response to the feedback, we have revised the manuscript accordingly. All modifications are highlighted in red in the revised version. Below, we summarize the key updates made specifically in response to the reviewer's comments.
> >
> > > Improve the clarity of 50% sparsity
> > To clarify that we report observed sparsity values, we have revised both the abstract and Table 1:
> >
> > - In the abstract, we now use "around 50%" instead of simply "50%"
> > - In Table 1, we now use "Measured Sparsity"
> >
> > > Strengthen the justification for exploring the co-design of hardware and algorithms
> >
> > We have revised Section 5.2 to provide a more comprehensive justification for hardware-algorithm co-design. The revisions include:
> >
> > - A detailed analysis of GPU energy consumption challenges and the critical need for specialized hardware
> > - An examination of how random weights reduce memory bandwidth requirements
> > - A comparative evaluation demonstrating the advantages of unstructured sparsity over structured approaches
> > - A quantitative assessment highlighting the benefits of native mixed-precision hardware support
> >
> > We are grateful to the reviewer's thoughtful and detailed feedback, which helped us improve the manuscript substantially.

---

### Review · Reviewer_hTvN · 2025-10-02

**Summary Of Contributions:**

The paper proposes a supermask method, called AdaSup, scalable for training LLMs and comparable to existing methods such as BitNet-1.58. The paper builds on top of the existing methods like EdgePopup and M-Sup (supports multivalue masks) and extend this for LLMs. Primarily, they replace the costly top-k operation in the computation of supermasks with a score quantization method. The base factor $\gamma$ in their quantization method is computed using the mean of the scores. Furthermore the method is extensible to n-bit multivalued masks by replacing the upper bound of the clip function by $2^n-1$. The paper then shows results for the propsed AdaSup outperforming other existing supermask methods (EdgePopup, Fixedthreshold) for pre-training (lower validation perplexity) and also comparable to EdgePopup masks (2-bit / 3-bit) for validation perplexity (slightly worse, but more efficient overall). Following this, they compare the method to BitNet - showcasing better TPP scaling over bitnet (which flattens at higher TPP values), and further show that 2-bit masks are enough for high quality (with sparsity + low-bitwidths), and also for parameter scaling, where they show that TicketLLM (the models trained on AdaSup) scale better than BitNet. Finally, they show how these methods compare to post-training quantization (PTQ) and quantization-aware training (QAT) methods. At low bitwidtds (2-bit), the method outperforms all PTQ methods and standard LLMQAT. However, there's a tradeoff with ParetoQ, where they have lower accuracy, but have higher sparsities than those methods. The paper finally outlines the challenges of their method (not deployable on everyday hardware and building on top of existing methods for this).

**Audience:**

Yes

**Audience Explanation:**

The method, while building on top of existing Supermask methods for their base, extends them to be scalable to LLMs, one of the biggest limitations these methods have today. Furthermore, they simplify the design, while integrating multi-valued masking and their findings of how to initialize networks etc. are inline with previous methods. The discuss the clear tradeoffs on how the method will be using existing hardware routines such as random number generation, while also being clear of the drawbacks that prevent the practical adoption of their method (deployable only on domain-specialized hardware). Overall, the paper has enough novel aspects, while not complicating the design to achieve the required results.

**Claims And Evidence:**

Yes

**Claims Explanation:**

*Scalability of masks*

- from using top-k to quantization for the mask computation
- expressivness per bit is higher: With randomness giving the signs {$\pm$ 1}, weights can focus on magnitude. This allocates more bits to the supermask over the sign.
- training efficiency of the method: show that model can be trained with 40 GPU hours vs other multivalued masks, which take 80 GPU hours

*Better quality at higher sparisty*

- the method results in ~50% sparsity from the masking pattern relative to other methods like bitnet which can get only ~30% sparsity at similar bitwidths (2-bit).
- the overall validation perplexity of the method is lower relative to bitnets and the model is scalable to higher TPPs

*Overall efficiency*

- Lower overhead of mask computation (on-the-fly) vs sorting / gating for EdgePopup
- Better scoring mechanism via quantization reduces overheads during training (~30% time spent in Edgepopup on computing scores for Linear layers)

*Empirical Results*

- For most of their claims (lower validation perplexity / better tpp scaling / better parameter scaling) - they show concrete numbers (including downstream evals where necessary) and also graphs with concise explanations
- For their claims comparing PTQ / QAT methods - they show downstream results on low bitwidths against their trained models and show favorable results. Where they don't outperform QAT methods, they provide reasonable explanations for the quality differences.

**Requested Changes:**

- In the abstract, will be nice to highlight the abbreviation for SLT in the first line after Strong Lottery Tickets. Not highlighted anywhere else, but used immediately in the next few lines. [Ease of reading for users]
- Several citations (for example, page 2: Fineweb-edu / page 3: SLT) are not done correctly. Please fix this properly throughout the paper [Ease of reading for users]
- For the graphs were results are very close for different methods (eg, Figure 4b), will be good to have smaller dots representing the datapoints for better readability
- As model size increases, bitnet eventually starts outperforming (not by a large margin, but still does) over TicketLLMs. Will be good to understand the authors' intuitions behind these results. What happens if you scale it even further, does the gap start widening due to the higher sparsity fraction in the TicketLLM models [Strengthen work]

---

> ### Author Response · Authors · 2025-10-13
> **Response to Reviewer hTvN**
>
> We thank the reviewer’s thoughtful and constructive comments. Based on the feedback provided, we have revised our manuscript accordingly. All changes are highlighted in red in the revised manuscript. Below, we address each of the reviewer's comments and describe the key updates made in response.
>
> > Clarification that that bitnet eventually starts outperforming (not by a large margin, but still does) over TicketLLMs.
>
> Thank you for the insightful question. We currently do not have a definitive explanation for the difference in parameter scaling between TicketLLM and BitNet, but we suspect the discrepancy may be influenced by sub-optimal hyperparameter settings as low-bit models are often  more sensitive to such choices. A more comprehensive investigation would be valuable to fully understand this phenomenon.
>
> Nevertheless, it is important to note that TicketLLM exhibits superior scaling efficiency when considering effective bit count as demonstrated in Figure 6 (a). This characteristic makes TicketLLM particularly preferable for deployment in resource-constrained environments, where computational efficiency is critical.
>
> > What happens if you scale it even further, does the gap start widening due to the higher sparsity fraction in the TicketLLM models.
>
> When we scale the parameters further, we don’t expect the performance gap between BitNet and TicketLLM to widen despite TicketLLM’s higher sparsity fraction. This is because both models performance with respect to the parameter count will be bounded by the full-precision LLaMA baseline. Indeed, our scaling curve in Figure 6 (b) predict that the gap between TicketLLM and LLaMA diminishes with increasing parameters, though its convergence is more gradual than BitNet’s one. This prediction supports our expectation in further performance scale.
>
> Importantly, TicketLLM's sparsity does not degrade this scaling behavior. Instead, when we account for the computational benefits of sparsity (i.e., effective model bit count), TicketLLM actually provides better performance-per-compute than BitNet, as shown in Figure 6(a). Thus, the higher sparsity fraction enhances efficiency without compromising the convergent scaling trend.
>
> ---
> Furthermore, based on the reviewer's valuable feedback, we have made the following key changes to strengthen our manuscript:
>
> > Highlight the abbreviation for SLT
>
> In the first sentence in abstract, we have added the abbreviation for SLT.
>
> > Correct the reference
>
> We have fixed the incorrect references in page 2, page 10, and appendix.
>
> > Improve the graph readability
>
> We have modified both marker size and marker shape in Figure 4, Figure 5, and Figure 6.
>
> > Expected performance in further scaling
>
> We have added the discussion of expected performance for further parameter scaling in parameter scaling section in Section 4.3.
>
> - “With further parameter scaling, the performance gap between BitNet and TicketLLM is expected to remain limited,  as both models' performances are expected to be bounded by the full-precision LLaMA baseline. Indeed, the loss gap between LLaMA and 2-bit TicketLLMs narrows with increasing parameters, potentially achieving comparable loss around 100B parameters. This expected convergence suggests that all models would exhibit comparable absolute performance at scale. However, TicketLLM's higher sparsity translates to superior computational efficiency,  maintaining its practical advantage in resource-constrained deployments.”
>
> We are grateful to the reviewer's thoughtful and detailed feedback, which helped us improve the manuscript substantially.

---

### Decision · Action_Editor_mERV · 2025-11-14

**Recommendation:** Accept as is

**Audience:**

Yes

**Audience Explanation:**

The reviewers unanimously agree that this paper will be of interest to some members of TMLR's audience thanks to its extensive experiments, novel and simple supermask estimation algorithm, and strong empirical results. The reviewers expressed some concern that the method in the paper has high overhead because it requires training from scratch for each model and that the resulting models are incompatible with modern GPU hardware due to the use of unstructured sparsity, but the consensus is that this paper constitutes a useful addition to the literature that other researchers will want to build on.

**Claims And Evidence:**

Yes

**Claims Explanation:**

This paper makes two main claims:
1. Quantization of score parameters, realized in a method named Ada-Sup, provides a way to scalably and efficiently optimize multi-value supermasks for Transformer-based models.
2. Applying Ada-Sup to Transformer models with random weights drawn from {-1, 1} leads to a family of low-bit, sparse models, called TicketLLM, that perform well over a range of effective model size and dataset size.

The reviewers unanimously agree that these claims are supported by clear and convincing evidence. Claim (1) is supported by analysis and experiments showing that Ada-Sup is capable of generating multi-valued supermasks, that Ada-Sup requires only a single training run on a model, and that Ada-Sup has lower overhead than many competing methods. Claim (2) is supported by experiments showing that TicketLLM achieves better C4 validation loss than BitNet b1.58 across a range of training dataset sizes and model sizes and comparable or better downstream task performance, that TicketLLM has better scaling properties than BitNet b1.58 at effective bit counts below 10GB, and that TicketLLM is competitive with a number of quantization methods on MobileLLMs, except for ParetoQ.